# Brain activity regulates loose coupling between mitochondrial and cytosolic Ca$^{2+}$ transients

Yuan Lin [1,2], Lin-Lin Li[3], Wei Nie[1], Xiaolei Liu[4], Avital Adler[2], Chi Xiao[3], Fujian Lu[1], Liping Wang[4], Hua Han[3], Xianhua Wang[1], Wen-Biao Gan[2]* & Heping Cheng[1]*

Mitochondrial calcium ($[Ca^{2+}]_{mito}$) dynamics plays vital roles in regulating fundamental cellular and organellar functions including bioenergetics. However, neuronal $[Ca^{2+}]_{mito}$ dynamics in vivo and its regulation by brain activity are largely unknown. By performing two-photon $Ca^{2+}$ imaging in the primary motor (M1) and visual cortexes (V1) of awake behaving mice, we find that discrete $[Ca^{2+}]_{mito}$ transients occur synchronously over somatic and dendritic mitochondrial network, and couple with cytosolic calcium ($[Ca^{2+}]_{cyto}$) transients in a probabilistic, rather than deterministic manner. The amplitude, duration, and frequency of $[Ca^{2+}]_{cyto}$ transients constitute important determinants of the coupling, and the coupling fidelity is greatly increased during treadmill running (in M1 neurons) and visual stimulation (in V1 neurons). Moreover, $Ca^{2+}$/calmodulin kinase II is mechanistically involved in modulating the dynamic coupling process. Thus, activity-dependent dynamic $[Ca^{2+}]_{mito}$-to-$[Ca^{2+}]_{cyto}$ coupling affords an important mechanism whereby $[Ca^{2+}]_{mito}$ decodes brain activity for the regulation of mitochondrial bioenergetics to meet fluctuating neuronal energy demands as well as for neuronal information processing.

[1] State Key Laboratory of Membrane Biology, Beijing Key Laboratory of Cardiometabolic Molecular Medicine, Peking-Tsinghua Center for Life Sciences, Institute of Molecular Medicine, Peking University, Beijing 100871, China. [2] Skirball Institute of Biomolecular Medicine, Department of Neuroscience and Physiology, Department of Anesthesiology, Neuroscience Institute, New York University School of Medicine, New York, NY 10016, USA. [3] National Laboratory of Pattern Recognition, Institute of Automation, Chinese Academy of Sciences, Beijing 100190, China. [4] The Brain Cognition and Brain Disease Institute, Shenzhen Institutes of Advanced Technology, Chinese Academy of Sciences, Shenzhen 518055, China. *email: Wenbiao.Gan@nyulangone.org; chengp@pku.edu.cn

As one of the most prominent neuronal organelles, mitochondria are ubiquitously found in cell body, dendrites, and axons, including synaptic terminals. They play multifunctional roles in bioenergetics, cell fate decision, aging, reactive oxygen species (ROS), and $Ca^{2+}$ signaling[1–4]. Many lines of evidence show that increases of cytosolic $Ca^{2+}$ ($[Ca^{2+}]_{cyto}$) concentration stimulate mitochondrial $Ca^{2+}$ ($[Ca^{2+}]_{mito}$) uptake, resulting in $[Ca^{2+}]_{mito}$ transients[5–8]. Such $[Ca^{2+}]_{mito}$ activity depends on mitochondrial $Ca^{2+}$ uniporter (MCU) and its accessory regulatory subunits[9–12]. $[Ca^{2+}]_{mito}$ dynamics is thought to be essential in regulating fundamental cellular and organellar functions, including energy metabolism, cellular differentiation, cell death, and redox signaling[2,13–15]. In addition, $[Ca^{2+}]_{mito}$ is critically involved in various neuronal-specific physiological and pathophysiological processes. For instance, it has been shown that $[Ca^{2+}]_{mito}$ modulates synaptic vesicle endocytosis in central nerve terminals[16]. During long-term potentiation (LTP) induction, elevated $[Ca^{2+}]_{mito}$ triggers a rapid burst of mitochondria fission in dendrites for LTP expression[17]. $[Ca^{2+}]_{mito}$ in presynaptic terminals has been implicated in synaptic transmission and can be disturbed by Amyloid β, leading to increasing $Ca^{2+}$ influx in a mouse model of Alzheimer's disease[18–21]. Lower $[Ca^{2+}]_{mito}$ loads occur early in Huntington's diseases (HD) in both patient tissues and the brain of HD transgenic mice, which increase neuronal susceptibility to apoptotic stimuli[22].

Despite these in vitro research progresses, neuronal $[Ca^{2+}]_{mito}$ dynamics and its physiological regulation in the brain of awake behaving mammalians remain largely unknown. In the present study, we simultaneously visualized $[Ca^{2+}]_{cyto}$ and $[Ca^{2+}]_{mito}$ dynamics in excitatory neurons in the primary motor cortex (M1) or primary visual cortex (V1) of behaving mice, using two-photon microscopy (2PM) in conjunction with genetically-encoded $Ca^{2+}$ indicators GCaMP6f[23] and jRGECO1a[24]. Our results reveal a probabilistic, yet spatially coordinated, coupling of $[Ca^{2+}]_{mito}$ to $[Ca^{2+}]_{cyto}$ transients in vivo, its physiological modulation by brain activity, and the role of $Ca^{2+}$/calmodulin kinase II (CaMKII) in enhancing the $[Ca^{2+}]_{mito}$-to-$[Ca^{2+}]_{cyto}$ coupling fidelity.

## Results

**Mitochondrial network in M1 pyramidal neurons.** The shape, length, and network connectivity of mitochondria are diverse and highly dynamic, depending on cell type, subcellular location[25], physiological activity, and environmental context. The maintenance of mitochondrial morphology is relevant to $[Ca^{2+}]_{mito}$ handling[26], and mitochondrial $Ca^{2+}$ influx can also alter mitochondrial morphology[17,27]. To delineate neuronal mitochondrial morphology in situ, we performed 3-D electron microscopy (EM) to reconstruct the mitochondrial structure[28,29] in dendrites (Layer 1, L1) and somas (Layer 2/3, L2/3) of M1 pyramidal neurons in the brains from adult wild type C57BL6 mice ($n = 3$). Mitochondria within 6 randomly selected dendrites and 5 somas were traced and analyzed. Serial EM images and 3D reconstruction revealed that the majority of mitochondria in dendrites exhibited an elongated linear structure with regular regions interconnected by thin tubules analogous to the mitochondrial nanotunnels reported previously[30–32] (Fig. 1a, b). This result suggests that essentially immobilized dendritic mitochondria may communicate with each other via the nanotunneling mechanism[30]. By contrast, mitochondria in somas were short ovoids dispersed throughout the cytoplasm with little physical interconnectivity (Fig. 1c, d). The length of dendritic mitochondria was $4.7 \pm 0.9\ \mu m$, and that of somatic mitochondria was $1.1 \pm 0.02\ \mu m$ ($n = 35$ mitochondria in 6 dendrites, and $n = 2465$ somatic mitochondria in 5 somas; Fig. 1e). The average volume density of somatic mitochondria was ~3 times lower than

that of dendritic mitochondria (Fig. 1f). These data provide a structural basis for the analysis of the $[Ca^{2+}]_{mito}$ response to neuronal activity in subcellular regions of pyramidal neurons (see below).

**Imaging of $[Ca^{2+}]_{mito}$ and $[Ca^{2+}]_{cyto}$ dynamics in vivo.** To measure $[Ca^{2+}]_{mito}$ dynamics in pyramidal neurons in vivo, we fused genetically-encoded $Ca^{2+}$ indicator GCaMP6f with the mitochondrial targeting sequence of human thioredoxin 2 (TXN2) for its mitochondrial targeting and retention. We packaged it into recombinant adeno-associated virus (AAV) with its expression under the control of the $Ca^{2+}$/calmodulin-dependent protein kinase type II subunit alpha (CaMKIIα) promoter (i.e., AAV-CaMKIIα-mito-GCaMP6f) (Supplementary Fig. 1a), in order to achieve specific expression in excitatory neurons[33,34]. In cultured neurons from neonatal mouse cortex, virally expressed mito-GCaMP6f fluorescence overlapped with the staining of tetramethylrhodamine methyl ester (TMRM), a mitochondrial membrane potential indicator (Supplementary Fig. 1b). In live animals, three weeks after viral infection in M1, immunohistochemical staining showed that GCaMP6f fluorescence co-localized with the mitochondrial outer-membrane marker TOM20 in pyramidal neurons (Supplementary Fig. 1c). These results indicate that GCaMP6f was successfully targeted to neuronal mitochondria in vitro and in vivo.

To simultaneously detect $[Ca^{2+}]_{cyto}$ and $[Ca^{2+}]_{mito}$ dynamics, we co-expressed red-shifted $Ca^{2+}$ indicator jRGECO1a to the cytosol driven by the human synapsin-1 (SYN1) promoter (i.e., AAV-hsyn-jRGECO1a). Three to four weeks after infection of AAV-CaMKIIα-mito-GCaMP6f and AAV-hsyn-jRGECO1a in L2/3 of M1, we performed two-photon, dual-color imaging of $[Ca^{2+}]_{mito}$ and $[Ca^{2+}]_{cyto}$ in somas (L2/3) and dendrites (L1), through glass cranial window in awake, head-fixed mice (Fig. 2a–d, Supplementary Movies 1, 2). We observed sporadic but sudden, stepwise transient elevation of mito-GCaMP6f fluorescence in both dendritic and somatic mitochondria of M1 neurons when mice were under resting conditions. Since pH sensitivity common to GFP-based indicators might interfere with $Ca^{2+}$ measurement, we expressed mito-pHtomato[35], which lacks the $Ca^{2+}$-sensing calmodulin moiety, in L2/3 pyramidal neurons. Using similar experimental protocols, we did not observe any analogous mitochondrial events (Supplementary Movie 3), suggesting that mito-GCaMP6f reflected mitochondrial matrix $Ca^{2+}$ rather than pH dynamics.

We found that individual GCaMP6f-reported $[Ca^{2+}]_{mito}$ transients rose abruptly, reaching peak amplitude within 2–3 s and lasting for several tens of seconds before eventually returning to the baseline. Their rate of occurrence was $0.03 \pm 0.01$ events per soma per min or $0.1 \pm 0.03$ events in dendrites per 1000 $\mu m^2$ in L1 per min. Based on their temporal kinetics, $[Ca^{2+}]_{mito}$ transients were categorized into three main types (Fig. 2e, f, Supplementary Fig. 2a): the regular type with a monotonic decline after the peak, the plateaued type with a stable elevation, and the staircase type with multiple upward steps. These three types of events occurred in roughly equal proportion and exhibited comparable initial amplitudes regardless of their kinetics (Supplementary Fig. 2). Interestingly, dendritic $[Ca^{2+}]_{mito}$ transients displayed a length (~15 $\mu m$) extending well beyond a single mitochondrion, and somatic $[Ca^{2+}]_{mito}$ transients encompassed the whole network of mitochondria within a soma (Supplementary Fig. 3), even though they were physically unconnected as revealed by the 3-dimensional EM reconstruction. Such cell-wide synchrony of $[Ca^{2+}]_{mito}$ transients suggests the presence of a global factor to signal a large number of individual mitochondria in a single neuron to activate all at once (see below).

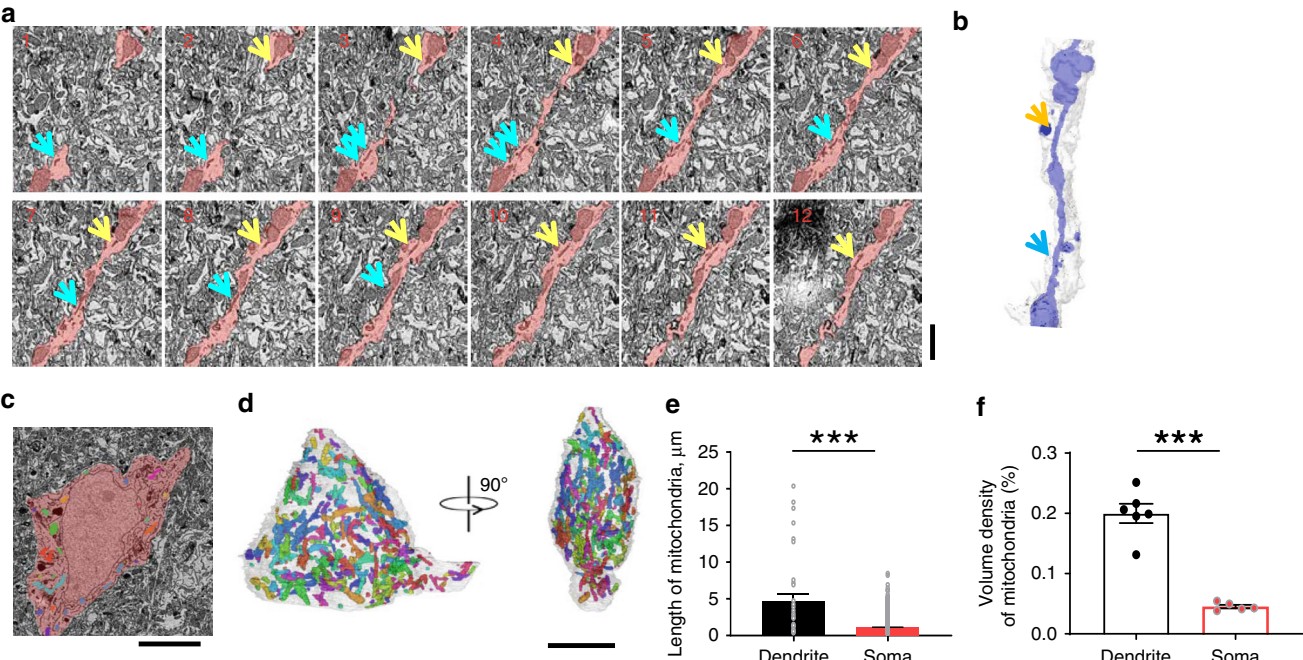

**Fig. 1** Mitochondrial morphology in the mouse M1 pyramidal neurons. Data were obtained with automated tape-collecting ultramicrotome scanning electron microscopy (ATUM-SEM). **a** Gallery view of SEM slices of the dendritic mitochondria. Yellow and green arrows indicate the mitochondria nanotunneling in the dendrite highlighted in pink. Scale bar: 1 μm. **b** Volume reconstruction of mitochondria (blue) in the dendrite (semi-transparent white) as shown by the pink area in (**a**). Yellow and green arrows indicate the mitochondria nanotunneling. **c** Image of a cell body of pyramidal neuron in M1. Scale bar: 5 μm. **d** 3D volume reconstruction of mitochondria in the mouse M1 neuron soma (semi-transparent white). $n = 291$ mitochondria identified are shown in different colors. Scale bar: 5 μm. **e** Statistics of mitochondrial length in dendrites and somas. $n = 35$ dendritic and 2465 somatic mitochondria. **f** Volume densities of mitochondria in dendrites and somas. $n = 6$ dendrites and 5 somas. Data are presented as mean ± SEM. *$P < 0.05$. **$P < 0.01$. ***$P < 0.0001$, unpaired $T$-test. Source data are provided as a Source Data file.

Parametric analysis revealed that $[Ca^{2+}]_{mito}$ and $[Ca^{2+}]_{cyto}$ transients were of distinctively different properties. The duration of $[Ca^{2+}]_{mito}$ transients was more than 10-fold longer than that of $[Ca^{2+}]_{cyto}$ transients: more than 75% of $[Ca^{2+}]_{mito}$ transients lasted 60 s or longer (Fig. 2h, i) while only less than 5% of $[Ca^{2+}]_{cyto}$ transients showed duration greater than 8 s. Furthermore, the frequency of $[Ca^{2+}]_{cyto}$ transients was ~10 times higher than that of $[Ca^{2+}]_{mito}$ transients both in dendrites and somas (Fig. 2j). The vast majority of $[Ca^{2+}]_{cyto}$ transients occurred without any stepwise increases of $[Ca^{2+}]_{mito}$. In contrast, every $[Ca^{2+}]_{mito}$ transient was triggered by an ongoing $[Ca^{2+}]_{cyto}$ transient, with its onset variably lagging behind the corresponding cytosolic event. Together, these results suggest a loose, probabilistic coupling of $[Ca^{2+}]_{mito}$ to $[Ca^{2+}]_{cyto}$ in vivo.

To quantify this unfaithful $[Ca^{2+}]_{mito}$-to-$[Ca^{2+}]_{cyto}$ coupling, we defined the coupling fidelity as the ratio between total numbers of $[Ca^{2+}]_{mito}$ and $[Ca^{2+}]_{cyto}$ transients ($N_{mito}/N_{cyto} \times 100\%$) (Fig. 3a). The average coupling fidelity was merely 3.3 ± 1.4% in L1 dendrites and 3.5 ± 1.3% in L2/3 somas (Fig. 3e). In both dendrites and somas, $[Ca^{2+}]_{cyto}$ transients in coupled events displayed higher peak amplitudes and greater durations compared to those in uncoupled events (Fig. 3b). Linear regression revealed a significant positive correlation between the amplitude pairs of coupled $[Ca^{2+}]_{cyto}$ and $[Ca^{2+}]_{mito}$ transients in somas, but not in dendrites (Fig. 3c, d).

**Brain activity enhances $[Ca^{2+}]_{mito}$-to-$[Ca^{2+}]_{cyto}$ coupling**. To investigate whether $[Ca^{2+}]_{mito}$ transients and particularly the $[Ca^{2+}]_{mito}$-to-$[Ca^{2+}]_{cyto}$ coupling are modulated by neuronal activity, we first imaged $Ca^{2+}$ activity in the primary motor cortex in mice running on a treadmill under the head-fixed condition (Fig. 4a)[36]. As shown in the heatmaps in Fig. 4b, c, the occurrence

of $[Ca^{2+}]_{cyto}$ transients in both L1 dendrites and L2/3 somas increased substantially in a subpopulation of neurons during the running session (50 s for dendrite recording and 100 s for soma recording), and returned towards their basal levels upon cessation of running (Fig. 4b, c). Representative time course plots show that robust, stepwise somatic $[Ca^{2+}]_{mito}$ transients were coupled to the much more frequent $[Ca^{2+}]_{cyto}$ transients, but with complex coupling and uncoupling patterns (Fig. 4; Supplementary Fig. 4). For instance, whereas enduring or bursting low-amplitude $[Ca^{2+}]_{cyto}$ transients could trigger $[Ca^{2+}]_{mito}$ transients, some large-amplitude, short-lived $[Ca^{2+}]_{cyto}$ transients failed to do so, and no single attribute (e.g., amplitude or duration) of $[Ca^{2+}]_{cyto}$ transient could solely determine the coupling (Supplementary Fig. 4). On average, the coupled $[Ca^{2+}]_{cyto}$ transients had higher peak amplitude and longer duration than those of uncoupled events (Fig. 4d), as was the case during the pre-run session (Fig. 3b). $[Ca^{2+}]_{cyto}$ transients in somas showed higher peak amplitudes and longer durations during running than at rest, while the duration rather than the peak amplitude of dendritic $[Ca^{2+}]_{cyto}$ transients increased during exercise (Fig. 4e–g). In both dendrites and somas, there were ~2-fold increases of $[Ca^{2+}]_{cyto}$ transient frequencies (Fig. 4h, i). Simultaneously recorded $[Ca^{2+}]_{mito}$ transients showed even more striking increases of frequency, whereas their amplitude and duration were comparable with and without exercise (Fig. 4e–i).

By quantifying the coupling fidelity in dendrites and somas in the pre-run and running sessions (Fig. 4j), we found that the average coupling fidelity during running was elevated to 19.3 ± 3.2 in dendrites and 13.9 ± 2.2 in somas, which were ~4 and ~3 times higher than their respective values at rest (Fig. 4j). Since L5 pyramidal neurons have distinct morphology, physiology and function in neural circuits as compared to L2/3 pyramidal

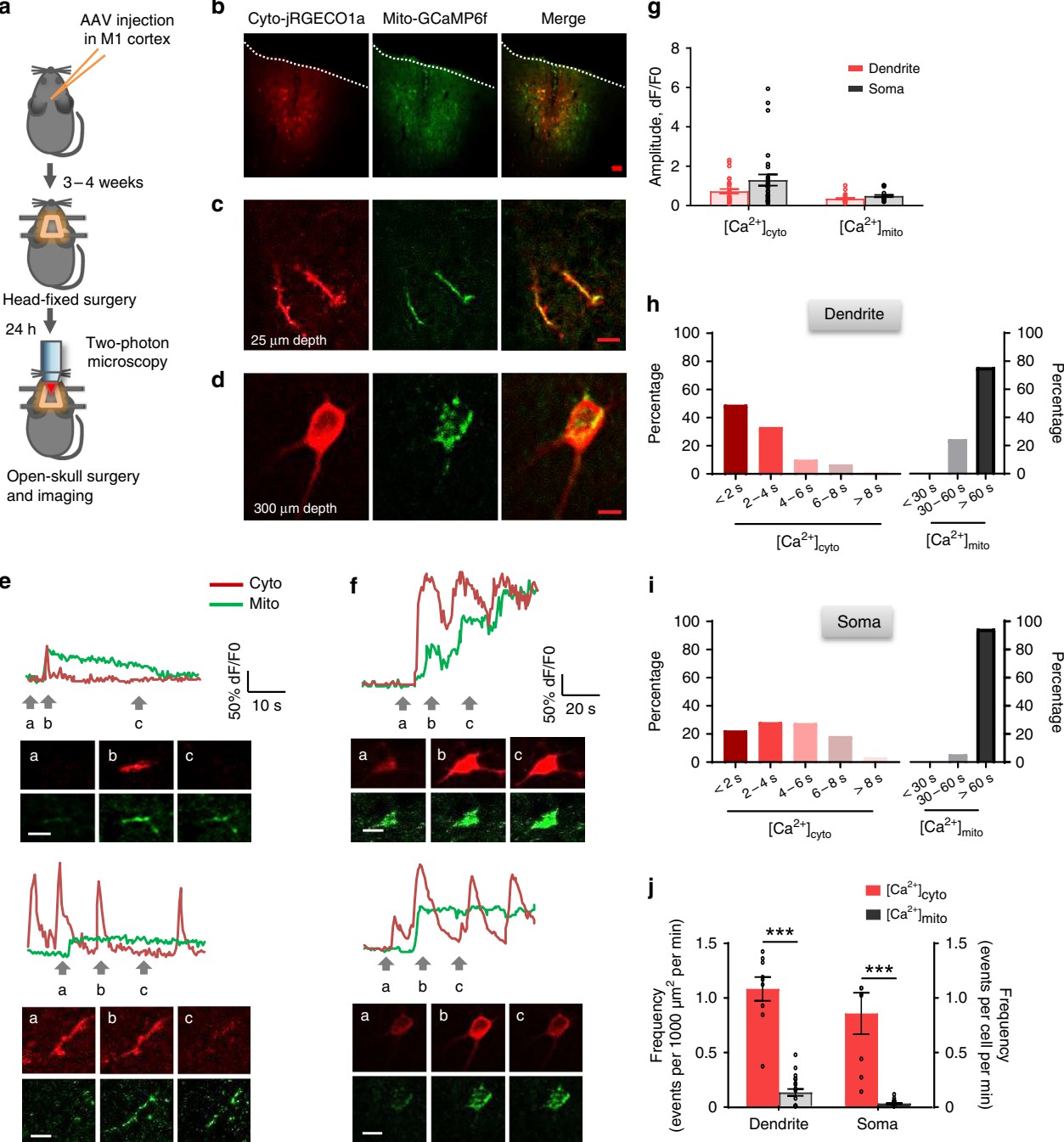

**Fig. 2** Imaging of $[Ca^{2+}]_{cyto}$ and $[Ca^{2+}]_{mito}$ transients in cortical neurons in awake mice. **a** Schematic showing experimental timeline for virus injection, head holder mounting, open-skull surgery, and two-photon imaging. Genetically-encoded $Ca^{2+}$ indicators, mito-GCaMP6f and jRGECO1a, were used to report cytosolic and mitochondrial $Ca^{2+}$ activity, respectively. **b** Confocal imaging of primary motor cortex neurons labeled by mtGCaMP6f and jRGECO1a in brain slice. Most of the labeled cells were in L2/3 (~150–350 μm deep from pial). Scale bars: 50 μm. **c, d** Representative 2PM images of double labeling of L1 dendrites at 25 μm depth (**c**) and a L2/3 soma at 300 μm depth (**d**). Scale bars: 20 μm. **e, f** Representative examples of pairs of $[Ca^{2+}]_{cyto}$ and $[Ca^{2+}]_{mito}$ transients in L1 dendrites (**e**) and L2/3 somas (**f**). Upper panels: time courses of fluorescence changes. Lower panels: corresponding dual channel images at three time points marked by arrows. Note that $[Ca^{2+}]_{mito}$ transients display highly variable kinetics that can be mostly categorized into regular (upper panel of **e**), plateaued (lower panels), and staircase groups (upper panel of **f**). Scale bars: 10 μm (**e**) and 20 μm (**f**). **g** Amplitudes of $[Ca^{2+}]_{cyto}$ and $[Ca^{2+}]_{mito}$ transients in dendrites ($n = 29$ $[Ca^{2+}]_{cyto}$ events and $n = 29$ $[Ca^{2+}]_{mito}$ events from 6 mice) and somas ($n = 29$ $[Ca^{2+}]_{cyto}$ events and $n = 22$ $[Ca^{2+}]_{mito}$ events from 8 mice). **h, i** Distribution of duration of $[Ca^{2+}]_{cyto}$ and $[Ca^{2+}]_{mito}$ transients in dendrites (**h**) and somas (**i**) (**h**: $n = 6$ mice; **i**: $n = 8$ mice.). **j** Frequencies of spontaneous $[Ca^{2+}]_{cyto}$ and $[Ca^{2+}]_{mito}$ transients in dendrites ($n = 9$ $[Ca^{2+}]_{cyto}$ events and $n = 21$ $[Ca^{2+}]_{mito}$ events from 6 mice) and somas ($n = 8$ $[Ca^{2+}]_{cyto}$ events and $n = 25$ $[Ca^{2+}]_{mito}$ events from 8 mice). Data are presented as mean ± SEM. *$P < 0.05$. **$P < 0.01$. ***$P < 0.0001$, unpaired $T$-test. Source data are provided as a Source Data file.

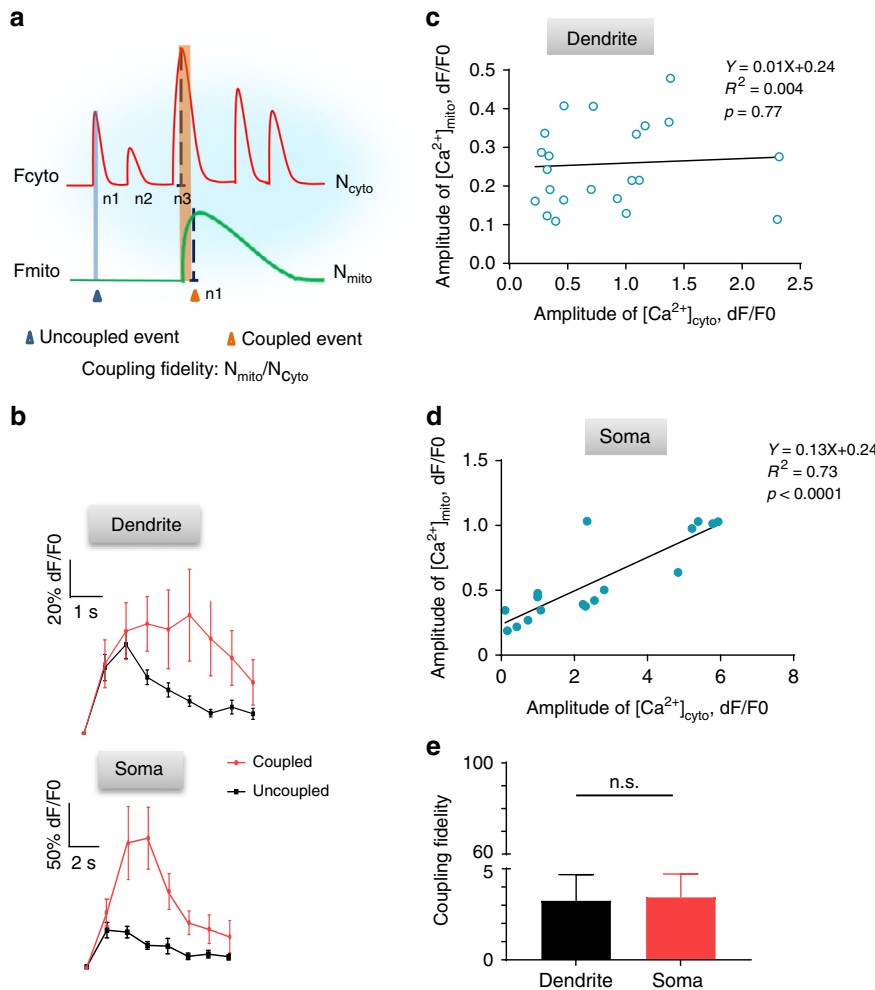

**Fig. 3** Dynamic coupling of $[Ca^{2+}]_{mito}$ to $[Ca^{2+}]_{cyto}$ transients in neurons in vivo. **a** Schematic showing $[Ca^{2+}]_{cyto}$ transients with or without coupled $[Ca^{2+}]_{mito}$ transients. $N_{cyto}$ or $N_{mito}$: number of cytosolic or mitochondrial $Ca^{2+}$ transients observed. Coupling fidelity is defined as $N_{mito}/N_{cyto} \times 100\%$.
**b** Averaged $[Ca^{2+}]_{cyto}$ transients in dendrites (upper panel) and somas (lower panel). 10–29 traces were used for each group. **c**, **d** Correlation of amplitudes of $[Ca^{2+}]_{cyto}$ and $[Ca^{2+}]_{mito}$ transients for coupled events in dendrites (**c**) and somas (**d**). (**c**: $n = 23$ paired events from 6 mice; **d**: $n = 17$ paired events from 8 mice). **e** Coupling fidelity in dendrites and somas ($n = 47$ from 6 mice in dendrites, $n = 23$ from 8 mice in soma). Data are presented as mean ± SEM. *$P < 0.05$. **$P < 0.01$. ***$P < 0.0001$, unpaired $T$-test. n.s.: no significance. Source data are provided as a Source Data file.

neurons[37–40], we also examined the coupling fidelity and its possible modulation by brain activity in somas and apical dendrites of L5 M1 neurons. Similar to L2/3 neurons, treadmill running enhanced frequencies and durations of $[Ca^{2+}]_{cyto}$ and $[Ca^{2+}]_{mito}$ transients, and augmented the $[Ca^{2+}]_{mito}$-to-$[Ca^{2+}]_{cyto}$ coupling fidelity by ~3.5 folds in dendrites and somas of L5 neurons (Supplementary Fig. 5). Further, we found that visual stimulation increased the coupling fidelities from $5.2 \pm 1.3$ to $11.3 \pm 1.6$ in somas and from $7.6 \pm 4.3$ to $25.8 \pm 3.4$ in apical dendrites of L2/3 pyramidal neurons in the primary visual cortex (Supplementary Fig. 6). Thus, physiological brain activity evoked by exercise or visual stimulation greatly enhances the $[Ca^{2+}]_{mito}$-to-$[Ca^{2+}]_{cyto}$ coupling fidelity in both dendrites and somas of M1 or V1 cortical neurons.

To evaluate the influence of mitochondrial function on the $[Ca^{2+}]_{mito}$-to-$[Ca^{2+}]_{cyto}$ coupling, we locally applied respiratory chain inhibitors, including rotenone, malonate, and oligomycin, and inhibitor of the mitochondrial permeability transition pore (mPTP) cyclosporine A in L2/3 neurons. We found that these respiratory inhibitors and the mPTP inhibitor all decreased the $[Ca^{2+}]_{mito}$-to-$[Ca^{2+}]_{cyto}$ coupling fidelity in L2/3 somas of M1 neurons, while their effects on the $[Ca^{2+}]_{mito}$ and $[Ca^{2+}]_{cyto}$

transients (amplitudes, duration, and frequency) were variable in an inhibitor-specific and context-sensitive manner (Supplementary Fig. 7).

By examining the time course of the coupled $[Ca^{2+}]_{mito}$ and $[Ca^{2+}]_{cyto}$ transients (Fig. 5a), we found that the onset of $[Ca^{2+}]_{mito}$ transients often coincided with the rise of $[Ca^{2+}]_{cyto}$ transients, and the coupling latency was typically less than 3 s in dendrites and 6 s in somas (Fig. 5b, c). By analyzing the temporal sequence of the dynamic coupling, we observed that high-frequency $[Ca^{2+}]_{cyto}$ activity tended to occur prior to a $[Ca^{2+}]_{mito}$ transient. In search for possible temporal determinants underlying the activation of a $[Ca^{2+}]_{mito}$ transient, we analyzed frequencies, amplitudes, and durations of $[Ca^{2+}]_{cyto}$ transients over eight 5-s time windows centered on $[Ca^{2+}]_{mito}$ transients (Fig. 5a). More numerous, brighter $[Ca^{2+}]_{cyto}$ transients were found within 5 s prior to an imminent $[Ca^{2+}]_{mito}$ transient in both dendrites and somas (Fig. 5d–g). These results suggest that the activation of $[Ca^{2+}]_{mito}$ transients depends also on the recent history of $[Ca^{2+}]_{cyto}$ activity. Interestingly, both the amplitudes and durations of somatic $[Ca^{2+}]_{cyto}$ transients were significantly increased after the coupling (Fig. 5f–i), suggesting a possible reciprocal effect of elevated $[Ca^{2+}]_{mito}$ on $[Ca^{2+}]_{cyto}$ dynamics.

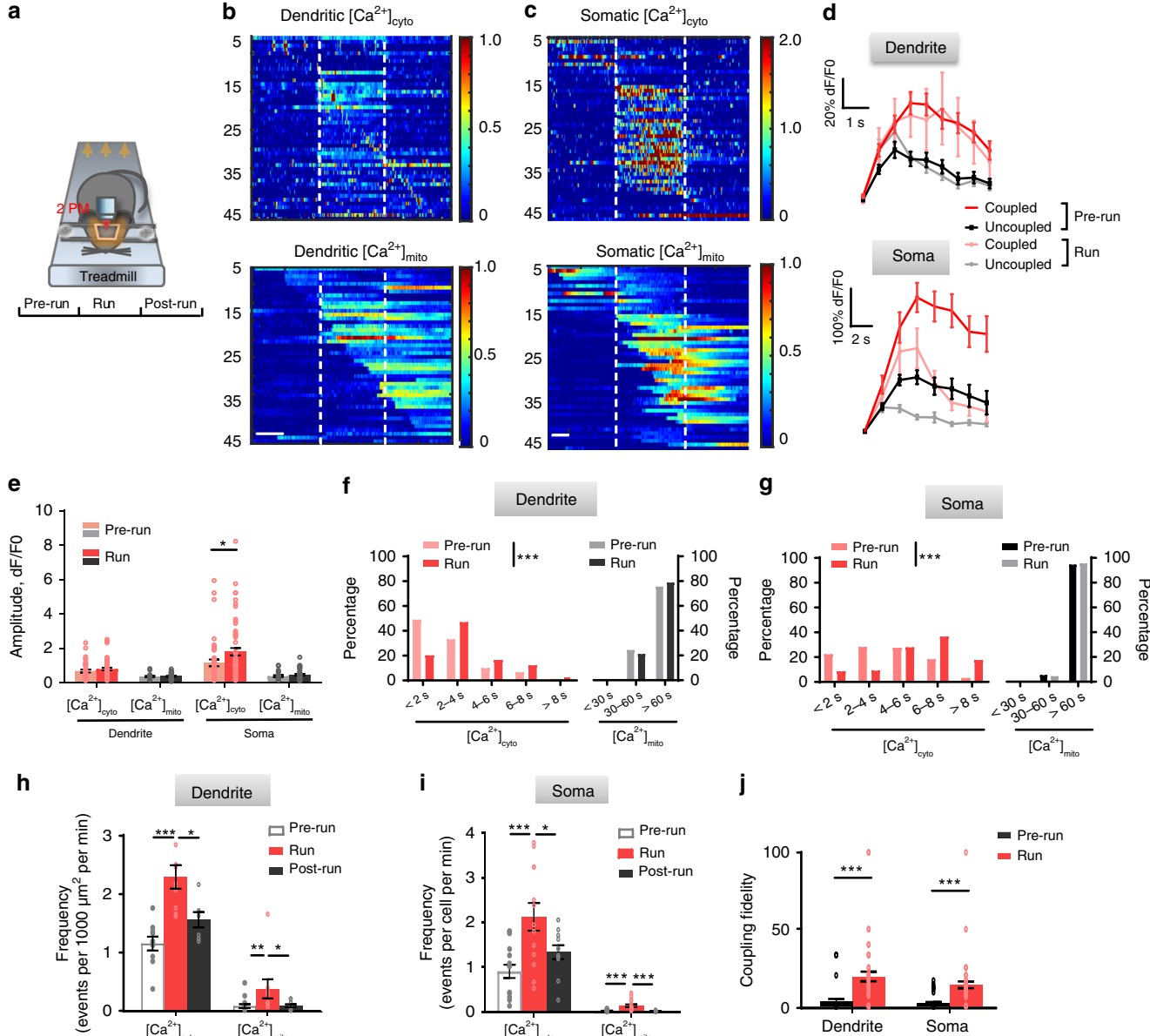

**Fig. 4** Enhanced $[Ca^{2+}]_{mito}$-to-$[Ca^{2+}]_{cyto}$ coupling amidst exercise-elicited neuronal activity. **a** Schematic of transcranial two-photon imaging in the forelimb motor cortex of awake, head-restrained mice before, during and after running on treadmill. **b**, **c** Heatmaps of $[Ca^{2+}]_{cyto}$ and $[Ca^{2+}]_{mito}$ activity in dendrites and somas under different conditions. Vertical dashed lines partition records obtained in the pre-run, run, and post-run stages. Each row represents $Ca^{2+}$ activity from a dendrite (**b**, $n = 48$ events) or a soma (**c**, $n = 49$ events). Pairs of $[Ca^{2+}]_{cyto}$ (upper panels) and $[Ca^{2+}]_{mito}$ activities (lower panels) were sorted by time of peak amplitude of $[Ca^{2+}]_{cyto}$ activity. Scale bars: 20 s. **d** Averaged $[Ca^{2+}]_{cyto}$ transients in dendrites (upper panel) and somas (lower panel) prior to and during treadmill running. 52–60 traces were used for each group. **e** Amplitudes of $[Ca^{2+}]_{cyto}$ and $[Ca^{2+}]_{mito}$ transients in dendrites ($n = 18$–55 dendrites from 6 mice) and somas ($n = 21$–60 somas from 7 mice) during pre-run and run stages. **f**, **g** Histogram of duration of $[Ca^{2+}]_{cyto}$ and $[Ca^{2+}]_{mito}$ transients in dendrites (**f**) and somas (**g**) during pre-run and run stages. (**f**: $n = 66$ dendrites from 6 mice; **g**: $n = 67$ somas from 7 mice. Mann–Whitney $U$-Test). **h**, **i** Exercise increased $[Ca^{2+}]_{cyto}$ frequency in dendrites (**h**, $n = 7$–26 from 6 mice) and somas (**i**, $n = 11$–17 from 7 mice). **j** Exercise increased $[Ca^{2+}]_{mito}$-to-$[Ca^{2+}]_{cyto}$ coupling fidelity ($n = 47$–49 dendrites and 47–48 somas from 6–7 mice,). Data are presented as mean ± SEM. *$P < 0.05$. **$P < 0.01$. ***$P < 0.0001$, by one-way ANOVA and multiple comparisons. Source data are provided as a Source Data file.

Taken together, our results suggest that both properties and recent history of $[Ca^{2+}]_{cyto}$ transients are important determinants of the $[Ca^{2+}]_{mito}$-to-$[Ca^{2+}]_{cyto}$ coupling.

Previous studies have shown that sudden $[Ca^{2+}]_{mito}$ transients are faithfully activated by $[Ca^{2+}]_{cyto}$ transients at ~100% coupling fidelity in mammalian cells both in vitro and in vivo using drug stimulation or field electrical stimulation at high frequency, respectively[8,41–46]. However, strong drug or electrical stimulation often elicit large $[Ca^{2+}]_{cyto}$ responses that may not represent physiological conditions. To better understand the variability of

$[Ca^{2+}]_{mito}$-to-$[Ca^{2+}]_{cyto}$ coupling, we performed in vitro experiments in cultured mouse cortical neurons with electrical filed stimulation at different frequencies (1–15 Hz, for 5 s). We found that robust $[Ca^{2+}]_{mito}$-to-$[Ca^{2+}]_{cyto}$ coupling occurred at 13 Hz or higher rate of stimulation, but became probabilistic at lower rates of stimulation (Supplementary Fig. 8a–d). The lower the electrical stimulation frequency was, the smaller the $[Ca^{2+}]_{cyto}$ transient was elicited, and the weaker the $[Ca^{2+}]_{mito}$-to-$[Ca^{2+}]_{cyto}$ coupling fidelity became in cultured neurons (Supplementary Fig. 8d). For coupled pairs of events, the onset of $[Ca^{2+}]_{mito}$ displayed a

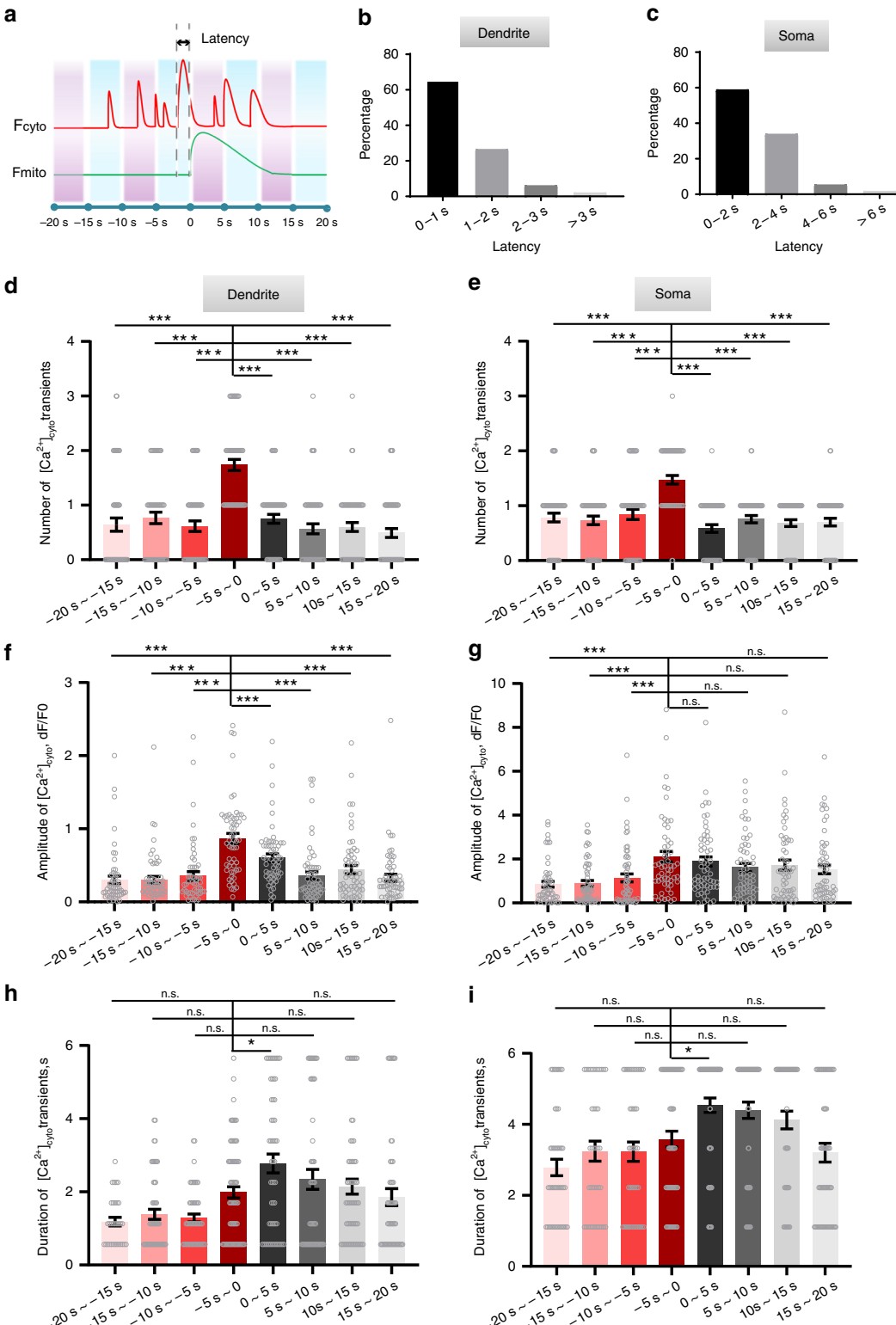

**Fig. 5** Temporal analysis of $[Ca^{2+}]_{mito}$ and $[Ca^{2+}]_{cyto}$ coupling. **a** Schematic of eight selected time windows centered on the onset of $[Ca^{2+}]_{mito}$ transient. Latency: the delay of the onset of a coupled $[Ca^{2+}]_{mito}$-to-$[Ca^{2+}]_{cyto}$ event. **b**, **c** Statistics of the coupling latency in dendrites (**b**, $n = 48$ events) and somas (**c**, $n = 49$ events). **d–i** Distribution of the number (**d**, **e**), peak amplitude (**f**, **g**), and duration of $[Ca^{2+}]_{cyto}$ transients (**h**, **i**) over different time windows in dendrites (**d**, **f**, **h**, $n = 42$–60 dendrites from 6 mice) and somas (**e**, **f**, **i**, $n = 40$–60 somas from 7 mice). Data are presented as mean ± SEM. *$P < 0.05$. **$P < 0.01$. ***$P < 0.0001$, by one-way ANOVA and multiple comparisons. n.s.:no signicance. Source data are provided as a Source Data file.

variable latency up to 6.3 s, with a mean value of 2.1 s (Supplementary Fig. 8c), and the $[Ca^{2+}]_{mito}$ amplitude positively correlated with its trigger $[Ca^{2+}]_{cyto}$ amplitude (Supplementary Fig. 8e). Thus, in vitro experiments reproduced many salient features of the $[Ca^{2+}]_{mito}$-to-$[Ca^{2+}]_{cyto}$ coupling in vivo. Moreover, the diversity of $[Ca^{2+}]_{mito}$-to-$[Ca^{2+}]_{cyto}$ coupling could be masked by the use of strong stimuli. As a preliminary observation, we also found that the rise of $[Ca^{2+}]_{mito}$ transient was accompanied by an increase in flavin adenine dinucleotide (FAD) autofluorescence (Supplementary Fig. 8f), suggesting that such a coupling significantly shifts mitochondrial energy and redox states.

**CaMKII regulation of $[Ca^{2+}]_{mito}$-to-$[Ca^{2+}]_{cyto}$ coupling.** In order to gain mechanistic insights into the coupling of $[Ca^{2+}]_{mito}$-$[Ca^{2+}]_{cyto}$, we examined the possible role of CaMKII, a multimeric $Ca^{2+}$-dependent kinase, since previous report has shown that CaMKII can decode both the intensity and frequency of $[Ca^{2+}]_{cyto}$ activity, and that its auto-phosphorylation endows its sustained activity over a time scale of seconds after $Ca^{2+}$-dependent activation[47]. Furthermore, CaMKII has been shown to affect MCU expression to regulate $[Ca^{2+}]_{mito}$ activity[48]. We, therefore, applied two CaMKII inhibitors (KN-62, KN-93, 100 μM), an inactive analog control (KN-92, 100 μM), and control artificial cerebrospinal fluid (ACSF) separately on the primary motor cortex and imaged the region of interest 20 min after.

We found that the frequency and amplitude of $[Ca^{2+}]_{cyto}$ transients were not significantly altered in either dendrites or somas by the CaMKII inhibitors during both pre-running and running stages (Fig. 6b–g). This result is in agreement with previous reports that CaMKII inhibitors block the downstream of $Ca^{2+}$ signaling pathway without affecting $[Ca^{2+}]_{cyto}$ activities in the pyramidal neurons in the motor cortex of awake mice[49,50]. Importantly, we found that the CaMKII inhibitors KN93 and KN62 not only reduced the frequency of $[Ca^{2+}]_{mito}$ transients when the animals were at rest, but also essentially prevented their responses to the high-rate $[Ca^{2+}]_{cyto}$ transients during treadmill running (Fig. 6d–i). By contrast, the inactive analog KN92 did not exert any significant effects on these sessions, confirming the specificity of the CaMKII inhibitors. Collectively, these results show that CaMKII is critically involved in regulating the fidelity of the $[Ca^{2+}]_{mito}$-to-$[Ca^{2+}]_{cyto}$ coupling.

## Discussion

Utilizing genetically-encoded $Ca^{2+}$ indicators that target different subcellular compartments, we revealed how $[Ca^{2+}]_{mito}$ is coupled to $[Ca^{2+}]_{cyto}$ in dendrites and somas of L2/3 excitatory neurons in the primary motor or visual cortex of awake behaving mice. Sudden, stepwise $[Ca^{2+}]_{mito}$ transients occurred in both dendrites (L1) and somas (L2/3, L5)) when mice were at rest or during motor or visual tasks. They arose with $[Ca^{2+}]_{cyto}$ transients acting as the trigger, and the overall $[Ca^{2+}]_{mito}$-to-$[Ca^{2+}]_{cyto}$ coupling fidelity was highly regulated by physiological neuronal activity. Furthermore, we provided evidence that CaMKII signaling plays a central role in regulating the coupling between $[Ca^{2+}]_{cyto}$ and $[Ca^{2+}]_{mito}$ transients.

Specifically, in M1 and V1 cortical neurons of awake behaving mice, we found that $[Ca^{2+}]_{cyto}$ transients are of variable amplitudes and frequencies, and that not every $[Ca^{2+}]_{cyto}$ transient triggers a companion $[Ca^{2+}]_{mito}$ transient. In fact, the fidelity of $[Ca^{2+}]_{mito}$-to-$[Ca^{2+}]_{cyto}$ coupling was merely 2–5% at rest, and rose to 11–26% during the treadmill running paradigm when the primary motor cortex was ~2-fold more active as judged by the $[Ca^{2+}]_{cyto}$ transient frequency. As such, the $[Ca^{2+}]_{mito}$-to-$[Ca^{2+}]_{cyto}$ coupling in pyramidal neurons occurs in a loose,

probabilistic rather than deterministic manner. Moreover, the coupling fidelity is modulated over a broad range by physiological neuronal activities and electrical stimulation, suggesting that $[Ca^{2+}]_{mito}$ signaling de-codes the intensity of brain activity.

For the quantification of $[Ca^{2+}]_{mito}$-to-$[Ca^{2+}]_{cyto}$ coupling, it is critical to ensure sensitive measurements for both $[Ca^{2+}]_{mito}$ and $[Ca^{2+}]_{cyto}$ events. The detection of $[Ca^{2+}]_{mito}$ response is challenged by multiple factors, including indicator affinity, non-linearity, and dynamic range (fluorescence for $Ca^{2+}$-free and bound species), as well as the basal and peak $Ca^{2+}$ levels, and mitochondrial fractional volume. We opted to use GCaMP6f and jRGECO1a, for dual-color measurement of $[Ca^{2+}]$ in mitochondria and cytoplasm because GCaMP6f exhibits high and comparable sensitivity in detecting $[Ca^{2+}]$ than jRGECO1a[23,24]. Even though alkalizing pH environment (~8.0 in mitochondrial matrix) would increase GFP fluorescence in a deprotonation-dependent and $Ca^{2+}$-independent manner[51], the dynamic range of GCaMP6f should be 3–4 times higher than that of jRGECO1a[24]. Despite technical limitations and uncertainties with the measurement of $[Ca^{2+}]$ in different cellular compartments, several lines of evidence suggest that the unfaithful $[Ca^{2+}]_{mito}$ coupling to $[Ca^{2+}]_{cyto}$ is likely a genuine physiological phenomenon. First, discrete $[Ca^{2+}]_{mito}$ transients, with their sudden and abrupt rises and distinctive long durations, were clearly discernible either from the baseline or on top of an ongoing $[Ca^{2+}]_{mito}$ transient. Even though $[Ca^{2+}]_{mito}$ transients were detected with a more sensitive indicator than $[Ca^{2+}]_{cyto}$, the frequency of $[Ca^{2+}]_{mito}$ events was much lower than that of $[Ca^{2+}]_{cyto}$. Second, CaMKII inhibition did not affect $[Ca^{2+}]_{cyto}$, but reduced the coupling fidelity between $[Ca^{2+}]_{cyto}$ and $[Ca^{2+}]_{mito}$, suggesting the involvement of CaMKII activity in the coupling process. Third, consistent with the involvement of biochemical signaling between $[Ca^{2+}]_{cyto}$ and $[Ca^{2+}]_{mito}$, conspicuous, yet highly variable latencies up to a few seconds were found in a significant portion of these coupled events. This observation further suggests that the onset of a $[Ca^{2+}]_{mito}$ transient could reflect the probabilistic gating of some fast mitochondrial $Ca^{2+}$ uptake mechanism (see below). In addition, our in vitro data show that the probabilistic nature of the $[Ca^{2+}]_{mito}$-to-$[Ca^{2+}]_{cyto}$ coupling in cultured neurons could be masked by strong electrical field stimulation used in previous studies[8,41–46].

Mitochondrial $Ca^{2+}$ uptake is dominated by the MCU-dependent $Ca^{2+}$ influx, which is regulated by multiple MCU components including MICU1, MICU2, and EMRE[9,11,52]. Only a certain range of $Ca^{2+}$ concentration in cytosol can activate MCU to uptake more $Ca^{2+}$ into mitochondria[14]. We therefore propose that the triggering of cell-wide $[Ca^{2+}]_{mito}$ transient reflects cooperative opening of MCUs in the mitochondrial network, via a $Ca^{2+}$/CaMKII-dependent mechanism. In this scenario, uncoupling between $[Ca^{2+}]_{mito}$ and $[Ca^{2+}]_{cyto}$ may reflect that MCUs are not in their open state. It is possible that the three types of $[Ca^{2+}]_{mito}$ kinetics reflect the patterns of interplay of several $Ca^{2+}$-regulatory mechanisms of the mitochondria, including the influx through MCU[46,53], the efflux through antiporter (mainly due to mitochondrial sodium calcium exchanger, NCLX)[54], and the flickering opening of the still elusive mPTP that acts as a fast $Ca^{2+}$ release channel[55]. For example, in coupled events, the regular kinetics of gradual decay of $[Ca^{2+}]_{mito}$ could be due to the efflux of mitochondrial $Ca^{2+}$ outweighing $Ca^{2+}$ influx shortly after MCU opening. The plateaued kinetics may be manifested when $Ca^{2+}$ influx rate equals $Ca^{2+}$ efflux rate or when both efflux and influx are shut off. The staircase kinetics may require both sustained activation of MCU and tandem occurrence of $[Ca^{2+}]_{cyto}$ transients.

Our analyses uncovered a few possible determinants for the coupling between $[Ca^{2+}]_{mito}$ and $[Ca^{2+}]_{cyto}$ transients. First and

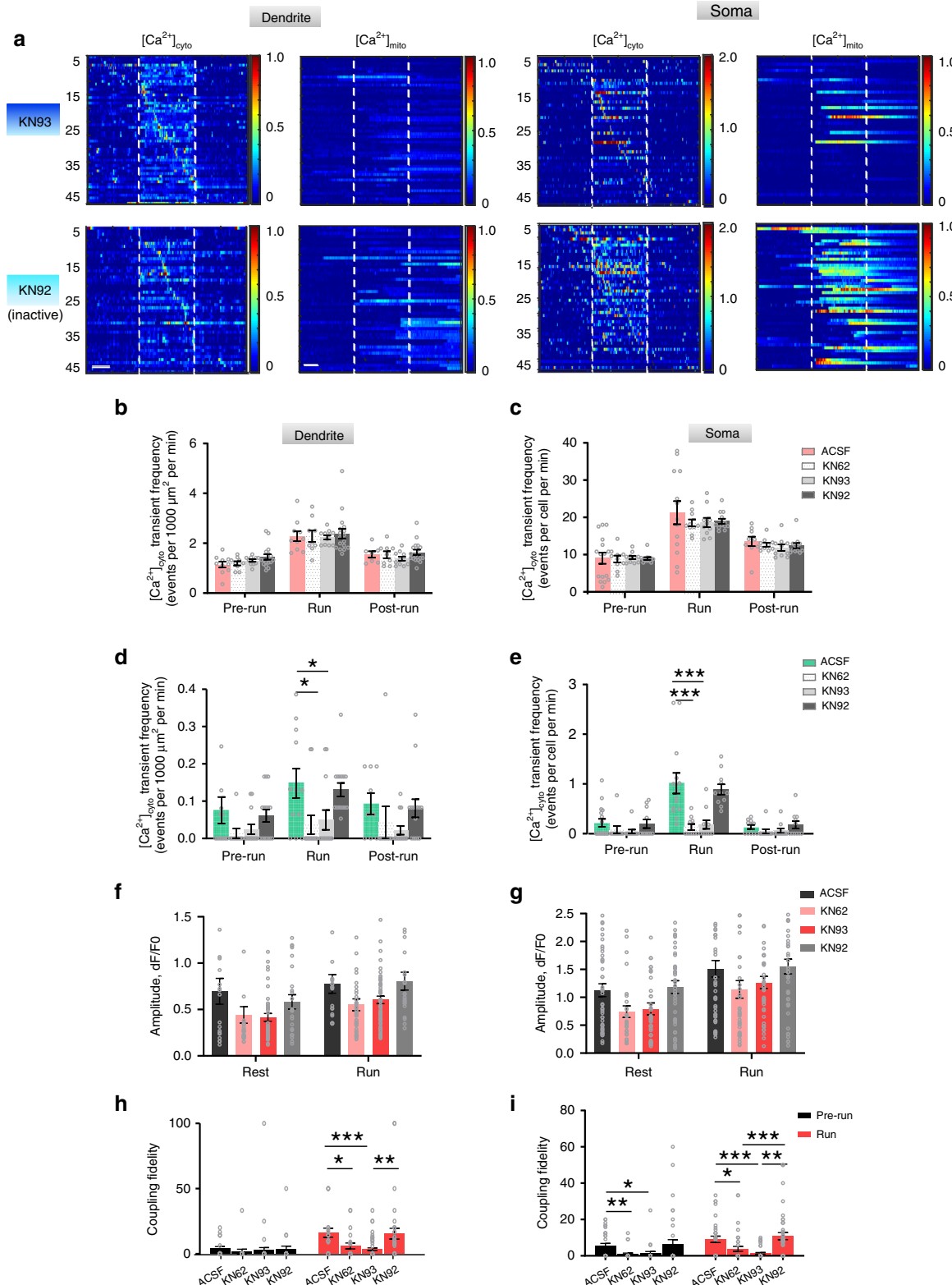

foremost, there seems to be a global cytosolic signal that, when at a supra-threshold level, permits the cell-wide mitochondrial network for the manifestation of $[Ca^{2+}]_{mito}$-to-$[Ca^{2+}]_{cyto}$ coupling. Second, the underlying permissive signal depends on the level of $[Ca^{2+}]_{cyto}$ since the coupled $[Ca^{2+}]_{cyto}$ transients were, on average, of greater amplitudes. In the staircase type of $[Ca^{2+}]_{mito}$ transients, we observed faithful coupling during successive $[Ca^{2+}]_{cyto}$ transients, suggesting the presence of a sustained supra-threshold signal that

allows for repetitive activation of the MCU mechanism. None-theless, any single attribute of $[Ca^{2+}]_{cyto}$ cannot be the sole deter-minant, as reflected by the complex pattern of the $[Ca^{2+}]_{mito}$-to-$[Ca^{2+}]_{cyto}$ coupling and uncoupling. In addition, a $[Ca^{2+}]_{cyto}$ frequency-dependent mechanism is likely involved: the $[Ca^{2+}]_{cyto}$ activity in a 5-s window prior to the onset was significantly higher than the average level. These salient features, including the coupling latency, the dependence of the coupling fidelity on the frequency of

**Fig. 6** Effects of CaMKII inhibition on $[Ca^{2+}]_{mito}$-to-$[Ca^{2+}]_{cyto}$ coupling. **a** Heatmaps of $[Ca^{2+}]_{cyto}$ and $[Ca^{2+}]_{mito}$ activities in dendrites and somas with KN93 and KN92 treatment. Vertical dashed lines partition records obtained in the pre-run, run, and post-run stages. Each row represents $Ca^{2+}$ activity from a dendrite (55 dendrites in KN93 group; 48 dendrites in KN92 group) or a soma (48 somas in KN93 group; 48 somas in KN92 group). Pairs of $[Ca^{2+}]_{cyto}$ and $[Ca^{2+}]_{mito}$ activities were sorted by time of maximum response of $[Ca^{2+}]_{cyto}$ activity. Scale bars: 20 s. **b–e** Effect of CaMKII inhibitors on $[Ca^{2+}]_{cyto}$ and $[Ca^{2+}]_{mito}$ transient frequency before, during and after running on treadmill in dendrites (**b**, **d**) and somas (**c**, **e**). $n = 4$–5 mice. 100 μM KN62, 100 μM KN93, 100 μM KN92 (inactive analog of KN93) in ACSF, or ACSF were administrated on the brain with flapped skull for 20 min before imaging. **f**, **g** Effect of CaMKII inhibitors on $[Ca^{2+}]cyto$ amplitude in dendrites (**f**) and somas (**g**) before and during running. $n = 4$–5 mice. **h**, **i** CaMKII inhibition decreased coupling fidelity in dendrites (**h**) and somas (**i**). Data are presented as mean ± SEM. *$P < 0.05$. **$P < 0.01$. ***$P < 0.0001$, by one-way ANOVA and multiple comparisons. Source data are provided as a Source Data file.

electrical stimulation, and the positive correlation between the amplitudes of $[Ca^{2+}]_{mito}$ and $[Ca^{2+}]_{cyto}$, were replicated in vitro. Thus, both attributes of ongoing $[Ca^{2+}]_{cyto}$ transient and recent history of $[Ca^{2+}]_{cyto}$ are determinants of the $[Ca^{2+}]_{mito}$-to-$[Ca^{2+}]_{cyto}$ coupling. Functionally, since mitochondrial $Ca^{2+}$ uptake stimulates energy metabolism through activating three key dehydrogenases of the tricarboxylic acid cycle[56], decoding of neuronal activity by $[Ca^{2+}]_{mito}$ signaling may provide a novel mechanism to satisfy the augmented ATP demand due to enhanced neuronal activity. Indeed, our preliminary experiments showed that the onset of $[Ca^{2+}]_{mito}$ transients may reciprocally elevate $[Ca^{2+}]_{cyto}$ signaling in vivo, and alter mitochondrial energy and redox states in cultured neurons.

CaMKII, as a multimeric $Ca^{2+}$-dependent kinase, is well-suited to integrate $[Ca^{2+}]_{cyto}$ to signal for $[Ca^{2+}]_{mito}$ activation. We provide direct evidence that CaMKII is central for permitting the $[Ca^{2+}]_{mito}$-to-$[Ca^{2+}]_{cyto}$ coupling and regulating its coupling fidelity. When we applied CaMKII inhibitors in brain in vivo[49,50], the $[Ca^{2+}]_{mito}$ transient frequency was markedly inhibited, and the activity-dependent enhancement of the coupling fidelity was also largely abolished, even though the $[Ca^{2+}]_{cyto}$ transient frequency per se and its increase by exercise were unaffected in either dendrites or somas. Therefore, we propose that a suprathreshold level of CaMKII activity serves as the global signal permissive for the $[Ca^{2+}]_{mito}$-to-$[Ca^{2+}]_{cyto}$ coupling. That the dynamic coupling depends on both $[Ca^{2+}]_{cyto}$ transient amplitude and frequency is also in good agreement with known properties of CaMKII activation, which decodes nonlinearly the amplitude and frequency of cytosolic $Ca^{2+}$ signal and retains a memory of recent $Ca^{2+}$ history on the order of a few seconds[57]. CaMKII could directly facilitate MCU activity via the phosphorylation mechanism[58]. Alternatively, the activation of CaMKII elevates sarcoplasmic reticulum $Ca^{2+}$ leak[59], which has been shown to contribute to mitochondrial $Ca^{2+}$ uptake[60] presumably via local $Ca^{2+}$ signaling at the mitochondrial-sarcoplasmic/ endoplasmic reticulum junctions[7,61]. Notably, it has been shown that the activation of CaMKII over a prolonged time scale can repress MCU expression as an early intermediate gene regulator to prevent excitotoxic cell death[48].

In summary, by investigating cytosolic and mitochondrial $Ca^{2+}$ dynamics and their coupling in the primary motor and visual cortical neurons of awake behaving mice, we show, for the first time, that the neuronal $[Ca^{2+}]_{mito}$-to-$[Ca^{2+}]_{cyto}$ coupling fidelity in vivo is very low at rest. More importantly, it is highly regulated by physiological neuronal activity. Mechanistically, CaMKII signaling plays a central role in regulating the dynamic mitochondrial response to cytosolic $Ca^{2+}$ signaling. That $[Ca^{2+}]_{mito}$ signaling could decode neuronal activity of different intensities and temporal patterns provide a natural mechanism for the mitochondrial participation of neuronal information processing and for the dynamic regulation of neuronal bioenergetics in keeping with brain activity. Future investigations are warranted to define the effectors downstream of CaMKII, and the mechanisms for spatial coordination of $[Ca^{2+}]_{mito}$ transients. Given the prominent and versatile roles of

mitochondrial $Ca^{2+}$ signaling in cell physiology and pathophysiology[15,62], it is also possible that dysregulation of the $[Ca^{2+}]_{mito}$-to-$[Ca^{2+}]_{cyto}$ coupling might underlie the initiation and progression of neuropsychiatric diseases from autism to Alzheimer's disease. As such, targeting the $[Ca^{2+}]_{mito}$-to-$[Ca^{2+}]_{cyto}$ coupling may provide a promising therapeutic strategy for combating these neuronal diseases.

## Methods

**Experimental animals**. Wild type C57BL6eij mice (without NNT deletion) were purchased from the Jackson Laboratory and housed in the animal facility at New York University School of Medicine. Mice were maintained at $22 \pm 2$ °C with a 12-h light: dark cycle (lights on 7 am, lights off 7 pm). Food and water were available ad libitum. 3-week old mice were used for AAV experiments and imaged 3–4 weeks later. Cortical neurons were cultured from postnatal 1-day-old wild type C57BL6eij mouse pups[63] (Beijing Vital River Laboratory Animal Technology Co. Ltd., Beijing, China). Preliminary observations were made in mito-GCaMP5 transgenic mice in C57BL6 background at Peking University. All experimental protocols were conducted in accordance with the institutional guidelines.

**Electron microscopy (EM) sample preparation**. Sample preparation was performed as followings[64,65]. Briefly, deeply anesthetized mice were perfused with 0.9% saline, followed by 4% paraformaldehyde (PFA, Sigma) and 2.5% glutaraldehyde (GA, Sigma) in PBS. The brains were dissected and post-fixed with 4% PFA and 2.5% GA at 4 °C overnight. On the second day, coronal slices ~200 μm thick were cut on a vibratome (Leica VT1000S). The M1 cortex region was dissected and post-fixed in 2% $OsO_4$ (Ted Pella) in 0.15 M sodium cacodylate (Biolink) buffer at room temperature (RT, 22–25 °C) for 90 min. Then the staining buffer was replaced by 2.5% ferrocyanide (Sigma) in 0.15 M sodium cacodylate buffer (pH 7.4) at RT for 1.5 h. The M1 cortex samples were washed with 0.15 M sodium cacodylate buffer and then incubated with filtered thiocarbohydrazide (TCH, Sigma) at 40 °C for 45 min, 2% $OsO_4$ at RT for 1.5 h, and 1% uranyl acetate (Merck) aqueous solution at 4 °C overnight. On the third day, the M1 cortex samples were washed with dd$H_2O$ at RT and then incubated in the lead aspartate solution, which contains 0.033 g lead nitrate (Sigma) in 5 ml 0.03 M aspartic acid (Sigma, pH 5.0), at 50 °C for 2 h. The brain sections were dehydrated through a graded ethanol series (50%, 70%, 80%, 90%, 100%, 10 min each) and pure acetone. Finally, the M1 cortex samples were embedded by epon-812 resin (SPI).

**EM images collection, alignment, and reconstruction**. The prepared M1 cortex samples were serially re-sectioned into 50 nm-thick ultrathin sections using an ATUMtome (Boeckeler Instruments, Inc., Tucson, USA) and were collected though Atlas 5 (Fibics incorporated, Ottawa, Canada) using SE with scanning EM (SEM, GeminiSEM300, Carl-Zeiss Microscopy GmbH, Oberkochen, Germany)[65,66]. For serial section images alignment, we utilized the non-linear image registration method which can guarantee the continuity of the reconstructed anatomical object across the sections[67]. In consideration of the different grayscale and contrast distributions, we adopted histogram equalization to weaken the noise and keep the consistency of raw images. To reconstruct mitochondria, dendrites and somas from EM images, we adopted an effective fully connected network to segment the mitochondria in 2-D electron microscopy images though deep learning method[67]. The network combined with Resnet50, PSPnet and Inception-like net[67]. Given an input image, we employed a pre-trained Resnet50 model which consisted of 4 residual blocks to extract the feature map. The first convolutional layers (kernel size $3 \times 3$ with a stride 2) in residual block 1 and 2 are used for down-sampling, while residual block 3 and 4 adopted dilated network strategy, which did not reduce image size. As a result, the output feature map size of Resnet50 model was only 1/8 of the input image. Next, the feature map was passed to the PSPnet. The inception-like structure was applied to up-sample and refined the per-pixel prediction. We also utilized a simple skip connection to conserve high-resolution feature information and propagate gradients back to the early layers. The training dataset consists of a stack of 6840 images with the resolution in 5 nm × 5 nm ×

50 nm. The testing dataset is with 437 slices in L1 area of the M1 cortex region and 332 slices in L2/3 area of the M1 cortex region. The number of pixels are 8928 × 8928 and 12288 × 12288, respectively. The adopted deep learning network was implemented by using Keras deep learning library and Tensor Flow backend. During training process, the network was optimized by Adaptive Moment Estimation (Adam) with the following optimization hyper parameters: learning rate = 0.0001, exponential decay rates for moment estimates $\beta\_1 = 0.9$, $\beta\_2 = 0.999$ and epsilon $= 10^{-8}$. Pixel-wise mean squared error was chosen as the loss function. In what follows, we applied Image J software to reconstruct the mitochondria. We imported the 2-D electron microscopy images which mitochondria were segmented into Image J software, and then displayed the 3-D reconstruction of mitochondria. Additionally, we utilized a 3-D connection method to determine connectivity of the mitochondrial network both in dendrites and somas.

**AAV reconstruction and packaging.** To create a mitochondria-targeted GCaMP6f (mito-GCaMP6f), the GCaMP6f sequence was first subcloned from pGP-CMV-GCaMP6f (Addgene, plasmid 40755) using primers 5'-ATTCGTTGG GGGATCCCCATGGTTCTCATCATCATC-3' (forward) and 5'-TTAACAACAA CAATTGTCACTTCGCTGTCATC-3' (reverse), and then in-frame fused with the mitochondrial target sequence of human TXN2 by inserting it into Mito-DsRed plasmid (gift from Dr. Quan Chen, Institute of Zoology, CAS) between BamHI and MfeI restriction sites. To obtain the AAV for expression of mito-GCaMP6f in vivo, mito-GCaMP6f was subcloned using primers 5'-GAGCGGTACCGGATCCTC TAGAGTGCGACTCCGGAGCCACCATGGCTCAGCGACTT-3' (forward) and 5'-GATTATCGATAAGCTTTTAACAACAACAATTG-3' (reverse), and inserted into pAAV-CaMK2α-hChR2(H134R)-EYFP (Addgene, #26969), replacing the hChR2(H134R)-EYFP sequence using BamHI and HindIII restriction sites.

The transfer plasmid, Rep/Cap, and the helper plasmid were co-transfected with pAAV- CaMKIIα-mito-GCaMP6f into HEK293 cells to produce AAV particles[68]. After two weeks of expression, AAVs were harvested and viral titration was done by quantifying vector genomes (vg) packaged into viral particles by quantitative PCR using primers 5'-GACTGAAGAGCAGATCGCAGAAT-3' (forward) and 5'-TGTCAGGAACTCAGGGAAGTCG-3' (reverse). The titer obtained was in the range of $10^{12}$ to $10^{13}$ vg ml$^{-1}$. Genetically-encoded calcium indicator jRGECO1a were expressed with recombinant AAV under the human synapsin-1 (SYN1) promoter (AAV, serotype 2/1; > $2 \times 10^{13}$ vg ml$^{-1}$; produced by the University of Pennsylvania Gene Therapy Program Vector Core).

**Two-photon Ca$^{2+}$ imaging in vivo.** For viral injection, AAV-CaMKIIα-mito-GCaMP6f and AAV-hsyn-jRGECO1a were mixed 1:1 in volume without dilution. A total of 0.1–0.2 µl of AAV viruses were injected (Picospritzer III; 18 p.s.i., 12 ms, 0.8 Hz) over 10–15 min into L2/3 or L5 of primary motor cortex or L2/3 of visual cortex using a glass microelectrode. Expression of Ca$^{2+}$ indicators was allowed for 3–4 weeks. 24 h before imaging, a head holder (two micro-metal bars) was attached on top of the skull with dental acrylic cement (Lang Dental Manufacturing Co.). An open-skull surgery was performed to put a glass window over the region of interest of the skull. The center of the region for Ca$^{2+}$ imaging is in the primary motor cortex (~0.5 mm anterior from bregma and ~1.2 mm lateral from the midline) or in the visual cortex (~3 mm anterior from bregma and ~3 mm lateral from the midline). During an imaging session of motor cortex, the head of the mouse was restrained by attaching the head holder to a custom stage fitted with free-floating treadmill[36,69,70]. During an imaging session of visual cortex, the head of the mouse was restrained by attaching the head holder to a custom stage fitted without free-floating treadmill. Mice were kept on the stage for a maximum of 1 h at a time. L2/3 somas of pyramidal neurons (positioned around 200–300 µm from pial surface), L5 somas of pyramidal neurons (positioned around 500–600 µm from pial surface) and their projected dendrites in L1 (positioned around 10–40 µm from pial surface) were imaged by using an Olympus Fluoview 1000 two-photon microscope (tuned to 1020 nm) equipped with a Ti:Sapphire laser (MaiTai DeepSee, Spectra Physics). Images were acquired with two GaAsP detectors with two-photon excitation at 1020 nm and dual-channel emission collection at 495–540 nm (for GCaMP6f) and 575–630 nm (for jRGECO1a). The average laser power on the tissue sample was 30–50 mW, and the frame rate was 2 and 1 Hz for imaging in L1 and L2/3 of the cortex respectively. All experiments were performed using a ×25 objective (numerical aperture 1.05) immersed in an ACSF solution and a ×2.0 (soma) or ×3.0 (dendrites) digital zoom. For the exercise protocol, the total imaging stack of each trial consisted of 300 frames, and animals were forced to run forward on the treadmill (speed increased gradually from 0 to 6 cm s$^{-1}$ within ~3 s) during the 101–200 frames. For the visual stimulation protocol, square-wave black/white and round-shaped drifting gratings (0.08 cycles per degree, 4 cycles per second, covering 32° × 32° screen area as seen by the mouse) of 8 changing orientations were used as visual stimuli. The total imaging stack of each trial consisted of 300 frames, and animals were subjected to visual stimuli during the 101–200 frames, while calcium imaging was performed simultaneously. The center of the drifting grating on monitor to perform stimuli is 18 cm away from the mouse's eyes, and refresh rate of the monitor is 75 Hz. Image acquisition was performed using FV10-ASW v.3.0 software and analyzed post hoc using NIH ImageJ software.

**Drug treatment.** ACSF containing 100 µM KN-93 (K1385, Sigma Aldrich), 100 µM KN-62 (I2142, Sigma Aldrich), 100 µM KN-92 (K112, Sigma Aldrich) was dumped on the surface of the imaging region on the primary motor cortex after removing the dura. After 20 min, a glass window was attached over the region of interest before imaging. In a previous study[49], we estimated that the final effective concentration for CaMKII inhibitors and dimethylsulphoxide (DMSO) were reduced to ~10 µM and <0.1%, respectively, due to diffusion. All of the drugs were dissolved in DMSO firstly to get stock solution and then dilute in ACSF to the final working concentration for each drug, with the final DMSO concentration lower than 1% (v/v).

**Cortical neuron culture, stimulation, and imaging.** The brain cortex was isolated and digested by 0.25% Trypsin-EDTA (Gibco) and washed twice in ice-cold modified HBSS without Ca$^{2+}$ and Mg$^{2+}$ (Thermo Fisher, Waltham, MA). The dissociated neurons were then placed onto poly-D-lysine (Sigma-Aldrich, MO) coated coverslips with density of 0.5–1 × 10$^6$ cells per ml. The cells are kept in neurobasal medium (Invitrogen) containing 2% B-27 supplement (Invitrogen), 1% GlutaMax (GIBCO), and 1% penicillin–streptomycin (GIBCO). The neurons were cultured at 37 °C in 5% CO$_2$. For mitochondrial Ca$^{2+}$ imaging, AAV-CaMKIIα-mito-GCaMP6f affection was performed on days in vitro (DIV) 7 for 5–7 days before imaging (0.5 µl virus with titer at 3.41 × 10$^{13}$ v.g. per ml was added in each well of 24-well plate) (Supplementary Fig. 8a–e). Or neurons were loaded with Rhod-2 AM (Invitrogen) prior to imaging, at 5 µM for 15 min and then washed for 3 × 5 min (Supplementary Fig. 8f). For cytosolic Ca$^{2+}$ imaging, neurons were loaded with Quest Rhod-4TM AM (AAT Bioquest) prior to imaging, at 5 µM for 15 min and then washed for 3 × 5 min. During imaging, cortical neurons were perfused in extracellular solution containing (in mM): NaCl 150, KCl 3, CaCl$_2$ 3, HEPES 10, MgCl$_2$ 1, glucose 8, pH 7.30, at the speed of 0.7 ml/min. Electrical field stimulation of about 8 V cm$^{-1}$ was applied through a pair of platinum wires, using 5 ms square-wave voltage pulses delivered from high current capacity stimulators. Neurons were stimulated with electrical pulses at different frequencies (1–15 Hz) for 5 s. Images were acquired with an inverted confocal microscope (Zeiss LSM NLO 710) equipped with an oil-immersion objective (Plan-Apochromat ×40, numerical aperture 1.3, Zeiss) by excitation at 488 nm (for GCaMP6f, FAD) and 543 nm (for Rhod2, Rhod4), and emission at 490–520 nm (for GCaMP6f), 495–540 nm (for FAD), 560–690 nm (for Rhod2) and 560–650 nm (for Rhod4), respectively at imaging rate of 1.28 Hz.

**Immunohistochemical staining.** Mice were deeply anesthetized with an intra-peritoneal injection of ketamine (100 mg per kg) and xylazine (15 mg per kg), and then perfused with 25 ml of 4% paraformaldehyde. Before slicing, brain was dissected out and post-fixed overnight in 4% paraformaldehyde. At the region of virus infection, coronal slices around 100 µm thick were cut on a vibratome by using Leica VT1000S. Fixed brain slices were rinsed with PBS three times and then permeabilized with 0.5% Triton X-100 and blocked with 10% normal goat serum. The slices were incubated with TOM20 (Rabbit, Santa Cruz, sc11415) overnight at 4 °C cold room. Next day, the slices were rinsed with PBS three times and then incubated with tetrarhodamine isothiocyanate (TRITC)-conjugated goat anti-rabbit IgG (Santa Cruz) in dark place for 1 h at room temperature. Slices were rinsed three times with PBS and then mounted between two glass coverslips in Vectashield (Vector Laboratories) and sealed with dental wax. The immunohistochemical staining slices were visualized using Zeiss LSM710 confocal microscope at 488 nm (Mito-GCaMP6f) and 568 nm (TRITC) excitation, and 490–550 nm and >560 nm emission, respectively.

**Two-photon and confocal microscopic imaging data analysis.** Regions of interests (ROIs) corresponding to visually identifiable double-labeling of GCaMP6f (in vivo two-photon imaging or in vitro confocal imaging) and Rhod-2 (in vitro confocal imaging) in mitochondria and jRGECO1a (in vivo two-photon imaging) or Rhod-4 (in vitro confocal imaging) in cytosol were selected for quantification. ROIs in green channel for imaging [Ca$^{2+}$]$_{mito}$ activity and the same ROIs in red channel for imaging [Ca$^{2+}$]$_{cyto}$ were analyzed. ImageJ plugin was used to register all imaging stacks. The fluorescence time course of each dendritic segment or cell body was measured with ImageJ software by averaging all pixels within the ROIs mentioned above. The fluorescence change was calculated as $\Delta F/F_0 = (F - F_0)/F_0 \times 100\%$, in which $F_0$ is the baseline fluorescence signal averaged over a 2-s period. The threshold for detecting a [Ca$^{2+}$]$_{cyto}$ or [Ca$^{2+}$]$_{mito}$ transient was set at more than three times the standard deviation of baseline fluorescence noise and duration greater than 1 s for mito-GCaMP6f, jRGECO1a, Rhod-2 and Rhod-4.

**Statistics.** Statistical analyses were performed using GraphPad Prism (version 7.0) and IBM SPSS 20. All datasets were tested for normality distribution first. For two group datasets, a two-tailed unpaired $t$-test was used when the dataset passed the normality test, otherwise a nonparametric test (Mann–Whitney test) was chosen. For more than two groups, one-way ANOVA analysis (either ordinary or Kruskal–Wallis) was performed, with a post hoc Dunn's test or Sidak's multiple comparisons test only if significant. Data are reported as mean ± SEM. A value of $P < 0.05$ was considered statistically significant.

**Reporting summary**. Further information on research design is available in the Nature Research Reporting Summary linked to this article.

## Data availability

The authors declare that all data supporting the findings of this study are available within the article and its Supplementary information files or from the corresponding authors upon reasonable request. The source data underlying Figs. 1e, f, 2g–j, 3c–e, 4e–j, 5b-i and 6b–i, and Supplementary Figs. 2a, c, d, 5b–i, 6b–i, 7a-g and 8c–e are provided as Source Data.

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

## Acknowledgements

We thank Dr. Jordi Magrane for providing the TOM20 antibody for immunostainings; Dr. Zhen Sun for critically reading the paper. This work was supported by National Key Basic Research Program of China (2017YFA0504000 and 2016YFA0500403), the National Science Foundation of China (31670039, 31970058, 8182780030, and 31821091) to H.C. and X.W., and NIH 5R01NS087198, 5R01NS047325 and 1R21AG061751 to W.-B.G.

## Author contributions

H.C., X.W., and W.-B.G. conceived the project. Y.L., W.-B.G., and H.C. designed research; Y.L. performed experiments, analyzed data, and made figures; Y.L. and F.L. did pilot experiments with mito-GCaMP5 transgenic mice; W.N. did the experiments in cultured neurons; L.L. and C.X. collected EM data under the supervision of H.H.; X.L. packaged AAV under the supervision of L.W.; A.A. ran MatLab for heatmap data sorting; Y.L., W.-B.G., and H.C. wrote the paper. All authors participated in data interpretation and discussion.

## Competing interests

The authors declare no competing interests.

## Additional information

**Supplementary information** is available for paper at https://doi.org/10.1038/s41467-019-13142-0.

