## [Peer Review File · Nature Communications]

Reviewers' Comments:

Reviewer #1:

Remarks to the Author:

In this paper, Lin and colleagues use *in vivo* two-photon imaging in mice to concomitantly record calcium transients in mitochondria and the cytosol of normal somata and dendrites in cortex. They find that mitochondrial calcium sometimes, but not always raises in response to cytosolic transients, that this coupling is increased with increased neuronal activity (induced by running) and, finally, that the coupling between cytosolic and mitochondrial calcium can be weakened by pharmacologically blocking CaMKII.

Overall, this methodologically sound paper is well written and easy to follow; but it is also a bit 'thin'. My main concern is the limited new insight that it contains and the impact that it will have. I also see a number of technical limitations that weaken the paper's significance further.

1) The paper's message is essentially that there seems to be threshold mechanisms for mitochondrial calcium to follow cytosolic calcium. Indeed, most of the data are consistent with this simple interpretation. The idea that in the field a 'hard' dogma of one-to-one coupling prevailed, irrespective of the kind of cytosolic signals, is not a fair representation of the dominant view in my opinion, but rather a bit of a strawman (see e.g. Devine & Kittler, Nat Rev Neurosci 2018 for discussion and more references). Given what we know about mitochondrial calcium entry (which is dominated by a Ca-sensitive Ca pore, MCU), the observed 'probabilistic' coupling is not surprising – it just shows that the parameters easily extracted from a cytosolic GECI fluorescence intensity are not sufficient to predict the mitochondrial calcium dynamics fully. The authors point out some deviations from this simple interpretation, especially that some substantial calcium transients are not followed by mitochondrial transients. However, here a technical limitation sets in – we do not really know the nature of the phenomena underlying the cytosolic transients; thus, just because two signals look similar does not mean they are the same. Without concomitant electrophysiology, this essentially remains hard to interpret – and the variability could entirely be a diversity in the different cytosolic calcium phenomena. We also do not know what 'absence of a transient' in mitochondria really means, as we have no calibration of sensitivity – it appears possible that in the much smaller volume sampled by the mitochondrially targeted sensor smaller transients are simply missed. Here, it also appears that a more quantitative measurement that also gives a measurement of baseline calcium could help (which a non-ratiometric intensity measurement does not and which could simply fluctuate in cytosol and mitochondria, with an impact on coupling efficiency), e.g. FRET or FLIM measurements. These would be hard, but potentially much more informative.

2) The effect of the CaMKII blockade is interesting, but its interpretation remains open (e.g. we do not know, how it affects the cell's or the circuit's physiology, the size and exact kinetics of the cytosolic signals, or the *in vivo* specificity of the drug) – these could be very remote effects. Overall, the paper stops short of testing some of the ideas that emerge from its observations – is the coupling having metabolic effects, or effects on any other known function of mitochondria? Does CaMKII e.g. change ER-mitochondrial coupling or proximity of mitochondria to the plasma membrane (or simply mitochondrial shape)? Does it affect the 'memory' of calcium transient 'history' that appears to influence how mitochondria react?

Minor:

3) Fig 1 is largely a repeat of known information that can be supplementary – relevant previous 3D EM work, e.g. of K. Harris should be cited.

4) Coupling potency does not make a lot of sense to me, as it is not clear what the range of this parameter and its linearity is. Two sensors are used here, with different, sigmoidal calcium response

curves that are not known in situ; so a simple ratio is not very meaningful (which suggests two essentially equivalent measurements that can simply be put in a relationship to each other, where '1' would mean both compartments reached the same calcium concentration).

Reviewer #2:

Remarks to the Author:

This paper addresses a highlight significant question about the association between cytoplasmic and mitochondrial calcium transients in neurons, in vivo. The paper is technically sound and makes a substantial contribution to the field of neuroscience and mitochondrial biology. The EM imaging data seems underutilized and is not so well connected with the rest of the data. Further analysis of the EM data or at least discussion in light of the functional data could strengthen the overall manuscript.

MAJOR comments

The study would be enhanced by evaluating the influence of mitochondrial function on the mitochondrial Ca²⁺ transients amplitude and frequency, and their coupling to cytoplasmic Ca²⁺ transients. Although this may be beyond the scope of the present paper, local application of respiratory chain inhibitors (Rotenone, Malonate, Oligomycin) or inhibitor of the mitochondrial permeability transition pore (mPTP – Nim811) would be useful to further understand the regulation of mitochondrial Ca²⁺ transient coupling, frequency, and amplitude.

Do the authors have any evidence to believe that the observed phenomenon is specific to the cell type studied or that it extends to other neurons and brain regions? Although the current data is of sufficient importance with these additional information, it would be quite meaningful to discuss in the manuscript.

MINOR comments

Line 75: this should be "mitochondrial nanotunnels" rather than "nanotubes". A comprehensive discussion of mitochondrial nanotunnels and comparison with tunneling nanotubes is available here at PMID 28935166.

Line 80: The authors state "the average volume of mitochondria" but the data seems to mitochondrial volume density (% of cell occupied by mitochondria). This point requires clarification. Also note of the percentage reported takes into account the volume of the whole cell body, or just the cytoplasm (cell body minus the nucleus).

Fig. S1: the colocalization images are generally sub-optimal. Nevertheless, it seems that there are some mitoGCaMP6f-positive structures that are TOM20-negative. And some TOM20-positive mitochondria that are mitoGCaMP6f-negative, particularly far away from the cell nucleus. These data seem to suggest that the probe is unlikely to label dendritic mitochondria far from the cell nucleus. Please comments on the uneven distribution of the probe and the implications for the interpretation of results.

Supplementary Fig S2: How large is the scale bar for microscopy images?

Figure 5G: x axis labels are shifted right.

The data in Figure 5 are intriguing. Why were 10s windows selected? Are the results maintained, or

possibly more striking, if 5 sec windows are used?

Please specify the full genotype of the mice. There are C57BL6j and C57BL6eij available at Jackson labs. The j version has a large-scale deletion in a gene encoding an important mitochondrial protein, NNT. This is not the case in the eij mice.

Line 336: For SEM images, 512 x 512 seems to refer to the number of pixels per image rather than the pixel size. The values given on line 338 are also unclear. Please correct and specify the resolution in nanometers.

Quantification of mitochondrial connectivity: please describe the method used with a sufficient level of details so the procedures could be independently replicated.

GENERAL REPLY

We sincerely thank both reviewers for positive evaluation and insightful comments of our manuscript. Your points are all well-taken, and the manuscript has been extensively revised with main improvements as the following:

1. We now presented data on $[Ca^{2+}]_{mito}$ -to- $[Ca^{2+}]_{cyto}$ coupling from Layer 5 (L5) neurons in the primary motor cortex (M1) (**Supplementary Figure 5**). Although L5 pyramidal neurons are distinctly different in morphology, physiology and function from L2/3 neurons, they also exhibit motor activity-dependent probabilistic $[Ca^{2+}]_{mito}$ -to- $[Ca^{2+}]_{cyto}$ coupling. These results complement and extend our conclusions drawn from L2/3 neurons in the motor cortex.
2. We also found probabilistic $[Ca^{2+}]_{mito}$ -to- $[Ca^{2+}]_{cyto}$ coupling in L2/3 neurons in the primary visual cortex under visual stimulation. The new results are reported in **Supplementary Figure 6**, and suggest that our conclusions are applicable to different brain regions and different modalities of brain activity.
3. We further investigated activity-dependent $[Ca^{2+}]_{mito}$ -to- $[Ca^{2+}]_{cyto}$ coupling in cultured cortical neurons. By varying the electrical field stimulation frequency, we identified stimulation frequency-dependence of the coupling fidelity, and the positive correlation between the amplitudes of the cytosolic and mitochondrial transients among the coupled events. These data provide further evidence that non-beat-to-beat coupling of $[Ca^{2+}]_{mito}$ -to- $[Ca^{2+}]_{cyto}$ is dependent on neuronal activity and are now presented in **Supplementary Figure 7**.
4. In addition, we have (1) re-analyzed data of **Figure 5** using 5-s, instead of 10-s, time windows; (2) omitted all contents related to “coupling potency” per the suggestion of Reviewer 1; (3) presented preliminary data in **Supplementary Figure 7**, showing a metabolic and redox effect of the $[Ca^{2+}]_{mito}$ -to- $[Ca^{2+}]_{cyto}$ coupling; (4) replaced the TOM20-mtGCaMP6f micrographs in **Supplementary Figure 1** with quality-improved data, and added new TMRM-mtGCaMP6f colocalization data; (5) presented more traces illustrating the complex patterns of $[Ca^{2+}]_{mito}$ -to- $[Ca^{2+}]_{cyto}$ coupling and uncoupling *in vivo* (**Supplementary Figure 4**); (6) described in full details the Methods for EM images collection, alignment, and reconstruction.

Reviewer #1 (Remarks to the Author):

In this paper, Lin and colleagues use *in vivo* two-photon imaging in mice to concomitantly record calcium transients in mitochondria and the cytosol of normal somata and dendrites in cortex. They find that mitochondrial calcium sometimes, but not always raises in response to cytosolic transients, that this coupling is increased with increased neuronal activity (induced by running) and, finally, that the coupling between cytosolic and mitochondrial calcium can be weakened by pharmacologically blocking CaMKII.

Overall, this methodologically sound paper is well written and easy to follow; but it is also a bit ‘thin’. My main concern is the limited new insight that it contains and the impact that it will have. I also see a number of technical limitations that weaken the paper's significance further.

We thank the reviewer for the positive appraisal of our manuscript. We have performed additional *in vivo* and *in vitro* experiments to address your suggestions and concerns. These new results are now integrated into the revised manuscript. We hope you will find this work is significantly improved.

1) The paper's message is essentially that there seems to be threshold mechanisms for mitochondrial calcium to follow cytosolic calcium. Indeed, most of the data are consistent with this simple interpretation. The idea that in the field a 'hard' dogma of one-to-one coupling prevailed, irrespective of the kind of cytosolic signals, is not a fair representation of the dominant view in my opinion, but rather a bit of a strawman (see e.g. Devine & Kittler, Nat Rev Neurosci 2018 for discussion and more references). Given what we know about mitochondrial calcium entry (which is dominated by a Ca-sensitive Ca pore, MCU), the observed 'probabilistic' coupling is not surprising – it just shows that the parameters easily extracted from a cytosolic GECI fluorescence intensity are not sufficient to predict the mitochondrial calcium dynamics fully. The authors point out some deviations from this simple interpretation, especially that some substantial calcium transients are not followed by mitochondrial transients. However, here a technical limitation sets in – we do not really know the nature of the phenomena underlying the cytosolic transients; thus, just because two signals look similar does not mean they are the same. Without concomitant electrophysiology, this essentially remains hard to interpret – and the variability could entirely be a diversity in the different cytosolic calcium phenomena. We also do not know what 'absence of a transient' in mitochondria really means, as we have no calibration of sensitivity – it appears possible that in the much smaller volume sampled by the mitochondrially targeted sensor smaller transients are simply missed. Here, it also appears that a more quantitative measurement that also gives a measurement of baseline calcium could help (which a non-ratiometric intensity measurement does not and which could simply fluctuate in cytosol and mitochondria, with an impact on coupling efficiency), e.g. FRET or FLIM measurements. These would be hard, but potentially much more informative.

The reviewer raised two important questions (1) the diversity of cytosolic calcium transients may explain why some of cytosolic calcium transients are coupled or uncoupled to mitochondria calcium transients, and (2) absence of coupling could be due to the use of non-ratiometric sensor which may not be able to detect small calcium transients in mitochondria.

Regarding the first question, we agree with the reviewer that the elevation of cytosolic calcium is mediated by a combination of multiple mechanisms (opening of calcium channels, ER release and calcium pump activity etc.). The coupling between $[Ca^{2+}]_{cyto}$ and $[Ca^{2+}]_{mito}$ is likely regulated by how $[Ca^{2+}]_{cyto}$ is elevated to trigger downstream CaMKII-dependent processes. Consistently, we found that whereas enduring or bursting low-amplitude $[Ca^{2+}]_{cyto}$ transients could trigger $[Ca^{2+}]_{mito}$ transients, some large-amplitude, short-lived $[Ca^{2+}]_{cyto}$ transients failed to do so. We found that no single attribute (e.g., amplitude or duration) of $[Ca^{2+}]_{cyto}$ transient could solely determine the coupling. Given the high variability of $[Ca^{2+}]_{cyto}$ events, which reflects the complexity of the underlying electrophysiological as well as neurochemical activities of a neuron, the manifestation of $[Ca^{2+}]_{mito}$ -to- $[Ca^{2+}]_{cyto}$ coupling

appears to be probabilistic, with a low coupling fidelity ranging from 2 to 20% in different brain regions responding to different modalities of physiological stimulation.

Nevertheless, we found that recent history of $[Ca^{2+}]_{cyto}$ transients constituted a significant determinant in this process (**Figure 5**). Further, we performed *in vitro* experiments and identified activity-dependent probabilistic $[Ca^{2+}]_{mito}$ -to- $[Ca^{2+}]_{cyto}$ coupling. In cultured mouse cortical neurons, we varied the frequency of electrical field stimulation over a 5-s episode. We found that, while $[Ca^{2+}]_{mito}$ -to- $[Ca^{2+}]_{cyto}$ coupling occurred in a robust, beat-to-beat manner at 13 Hz or higher rate of stimulation, it became probabilistic at lower rates of stimulation (**Supplementary Figure 7a-d**). The lower the electrical stimulation frequency applied, the smaller the $[Ca^{2+}]_{cyto}$ transient became and the weaker the $[Ca^{2+}]_{mito}$ -to- $[Ca^{2+}]_{cyto}$ coupling fidelity was in cultured neurons (**Supplementary Figure 7d**). For coupled pairs of events, the onset of $[Ca^{2+}]_{mito}$ displayed a latency up to 6.3 s, with a mean value of ~ 2.1 s (**Supplementary Figure 7c**); the $[Ca^{2+}]_{mito}$ amplitude positively correlated with its trigger $[Ca^{2+}]_{cyto}$ amplitude (**Supplementary Figure 7e**). These *in vitro* experiments reproduced many salient features of the $[Ca^{2+}]_{mito}$ -to- $[Ca^{2+}]_{cyto}$ coupling *in vivo* and suggest that the coupling is dependent on the level of neuronal activity.

Regarding the second question, we agree with the reviewer that the absence of coupling could be in part due to the use of non-ratiometric sensor, which may not be able to detect small transients (especially when neuronal activity is low). While the indicator GCaMP6f used for is non-ratiometric and unable to quantitatively measure the absolute $[Ca^{2+}]_{mito}$, it was one of the most sensitive Ca^{2+} sensors at the time of our experiments. We could clearly discern the onsets of sudden, discrete, stepwise $[Ca^{2+}]_{mito}$ transients, rising either from the baseline level or on top of an ongoing $[Ca^{2+}]_{mito}$ transient. Importantly, our findings (both *in vivo* and *in vitro*) strongly suggest that the coupling is dependent on recent history of cytosolic calcium, CaMKII activity and the level of neuronal activity. When neuronal activity is low, the high degree of polymorphism of $[Ca^{2+}]_{cyto}$ transients *in vivo* and the nonlinear, complex “computation” may confer the probabilistic nature of the $[Ca^{2+}]_{mito}$ -to- $[Ca^{2+}]_{cyto}$ coupling. As the reviewer pointed out, a more quantitative measurement of baseline calcium with FRET or FLIM could potentially be much more informative to understand the coupling. Since such experiments would require the development of more sophisticated sensors and detection methods, we hope the reviewer would agree that they are beyond the scope of the present study.

2) The effect of the CaMKII blockade is interesting, but its interpretation remains open (e.g. we do not know, how it affects the cell's or the circuit's physiology, the size and exact kinetics of the cytosolic signals, or the *in vivo* specificity of the drug) – these could be very remote effects. Overall, the paper stops short of testing some of the ideas that emerge from its observations – is the coupling having metabolic effects, or effects on any other known function of mitochondria? Does CaMKII e.g. change ER-mitochondrial coupling or proximity of mitochondria to the plasma membrane (or simply mitochondrial shape)? Does it affect the 'memory' of calcium transient 'history' that appears to influence how mitochondria react?

We thank the reviewer for these insightful comments. Our previous work have shown that blockade of CaMKII with KN93 prevented synaptic plasticity but did not affect the frequency and amplitude of dendritic calcium activity in layer 5 pyramidal neurons in the motor cortex¹. Consistently, we have also found that CaMKII blockade did not alter the size and kinetics of the cytosolic signals (**Figure 6a-g**). As for the possible functional significance of the coupling, we have made two preliminary observations. First, we showed that $[Ca^{2+}]_{cyto}$ transient amplitude as well as its duration was elevated after the onset of $[Ca^{2+}]_{mito}$ transients, suggestive of a reciprocal effect of $[Ca^{2+}]_{mito}$ elevation on cytosolic Ca^{2+} signaling (**Figure 5g**). Second, our *in vitro* data showed mitochondrial FAD autofluorescence was altered concurrently with $[Ca^{2+}]_{mito}$ transient (**Supplementary Figure 7f**), suggestive of a metabolic and bioenergetic effect of the coupling. Further, we did not observe any discernible change in mitochondrial morphology in the presence or absence of CaMKII inhibitors. At the present, we do not know if CaMKII e.g. change ER-mitochondrial coupling or proximity of mitochondria to the plasma membrane or if it affects the 'memory' of calcium transient 'history' that influences how mitochondria react. We hope the reviewer would agree that these are important mechanistic questions that could be addressed in the future work.

Minor: 3) Fig 1 is largely a repeat of known information that can be supplementary – relevant previous 3D EM work, e.g. of K. Harris should be cited.

Thank you for directing us to these 3D EM papers. Please see citation #24, #28, #29 and #30. The original aim of our 3D EM study was to address the puzzling observation that, despite a low coupling fidelity, $[Ca^{2+}]_{mito}$ transients was cell-wide synchronous. We sought to determine whether or not the synchrony arises from physical interconnectivity of the mitochondrial network. The major outcome of the structural data suggests that the spatial coordination is unlikely to be due to the physical interconnectivity of the mitochondrial network. Instead, other global factors (calcium and CaMKII) might be at work to synchronize individual mitochondria in the genesis of $[Ca^{2+}]_{mito}$ transients. To our

knowledge, that dendritic mitochondria may communicate via nanotunneling in our study was a novel finding that extends previous work on intermitochondrial communication. We hope the reviewer would be fine with our keeping the figure 1 in the main text.

4) Coupling potency does not make a lot of sense to me, as it is not clear what the range of this parameter and its linearity is. Two sensors are used here, with different, sigmoidal calcium response curves that are not known in situ; so a simple ratio is not very meaningful (which suggests two essentially equivalent measurements that can simply be put in a relationship to each other, where '1' would mean both compartments reached the same calcium concentration).

We agree with the reviewer and have now removed the term of “coupling potency” and pertinent contents (definition, figure panels, and text) from the manuscript. We sincerely thank the reviewer for all the insightful comments which help the improvement of our work.

Reviewer #2 (Remarks to the Author):

This paper addresses a highlight significant question about the association between cytoplasmic and mitochondrial calcium transients in neurons, in vivo. The paper is technically sound and makes a substantial contribution to the field of neuroscience and mitochondrial biology. The EM imaging data seems underutilized and is not so well connected with the rest of the data. Further analysis of the EM data or at least discussion in light of the functional data could strengthen the overall manuscript.

MAJOR comments

The study would be enhanced by evaluating the influence of mitochondrial function on the mitochondrial Ca²⁺ transients' amplitude and frequency, and their coupling to cytoplasmic Ca²⁺ transients. Although this may be beyond the scope of the present paper, local application of respiratory chain inhibitors (Rotenone, Malonate, Oligomycin) or inhibitor of the mitochondrial permeability transition pore (mPTP – Nim811) would be useful to further understand the regulation of mitochondrial Ca²⁺ transient coupling, frequency, and amplitude.

We thank the reviewer for the suggestions and have performed new experiments accordingly. As shown in the **Attached Figure** below, the overall conclusion is that inhibition of mitochondrial function by locally administered respiratory inhibitors as well as the MPTP inhibitor cyclosporine A all decreased the $[Ca^{2+}]_{mito}$ -to- $[Ca^{2+}]_{cyto}$ coupling fidelity in L2/3 somas of M1 neurons. Their effects on the $[Ca^{2+}]_{mito}$ and $[Ca^{2+}]_{cyto}$ transients (amplitudes, duration and frequency) were variable in an inhibitor-specific and context-sensitive manner. We would be glad to include these data in the manuscript upon the recommendation from the reviewer.

Do the authors have any evidence to believe that the observed phenomenon is specific to the cell type studied or that it extends to other neurons and brain regions? Although the current data is of sufficient importance with these additional information, it would be quite meaningful to discuss in the manuscript.

Concerning the general applicability of the findings in L2/3 neurons of the primary motor cortex, we now included new data to demonstrate similar results obtained in L5 M1 neurons (**Supplementary Figure 5**) and L2/3 neurons in the primary visual cortex modulated by visual stimulation (**Supplementary Figure 6**). These new data extended our conclusions to different types of cortical neurons responding to physiological stimulation of different modalities.

MINOR comments

Line 75: this should be “mitochondrial nanotunnels” rather than “nanotubes”. A comprehensive discussion of mitochondrial nanotunnels and comparison with tunneling nanotubes is available here at PMID 28935166.

We now corrected the term per your suggestion. Thank you.

Line 80: The authors state “the average volume of mitochondria” but the data seems to mitochondrial volume density (% of cell occupied by mitochondria). This point requires clarification. Also note of the percentage reported takes into account the volume of the whole cell body, or just the cytoplasm (cell body minus the nucleus).

Thank you for pointing this out. We now clarified that it referred to “mitochondrial volume density”, and the volume was inclusive of the nucleus.

Fig. S1: the colocalization images are generally sub-optimal. Nevertheless, it seems that there are some mitoGCaMP6f-positive structures that are TOM20-negative. And some TOM20-positive mitochondria that are mitoGCaMP6f-negative, particularly far away from the cell nucleus. These data seem to suggest that the probe is unlikely to label dendritic mitochondria far from the cell nucleus. Please comments on the uneven distribution of the probe and the implications for the interpretation of results.

The reason why some TOM20-positive mitochondria are mitoGCaMP6f- negative is because mitoGCaMP6f only labeled those AAV-infected excitatory neurons, while TOM20 could stain all cells. Some of the sparse mito-GCaMP6f-positive structure looks like TOM20-negative in images. However, we could see the faint signal in red-channel when zoomed in. The reviewer is correct that the probe did not label only dendritic mitochondria. In some cases, based on those mitochondria position and their short ovoid fragments dispersed structure, they are probably mitochondria in axons^{2,3}. Furthermore, the uneven distribution could be caused by the possibility of relatively lower expression of TOM20 in mitochondria in axon.

In the revised manuscript, we repeated the TOM20 immunostaining and chose better colocalization images instead of previous one (**revised supplementary figure 1c**). We also included data from cultured cortical neurons showing colocalization of mitoGCaMP6f with TMRM, an indicator of mitochondrial membrane potential which was also widely used here as a mitochondrial marker (**revised supplementary figure 1b**).

Supplementary Fig S2: How large is the scale bar for microscopy images?

We clarified that the scale bar is 10 μm . Thank you for pointing out this oversight.

Figure 5G: x axis labels are shifted right.

It is now re-aligned in the revised figure.

The data in Figure 5 are intriguing. Why were 10s windows selected? Are the results maintained, or possibly more striking, if 5 sec windows are used?

Considering that the average duration of $[Ca^{2+}]_{cyto}$ transients was around 3s, we chose to use the 10-s time window. Per your suggestion, we re-analyzed the data using 5-s windows, and revealed similar, but finer, results which are reported in the **revised Figure 5**.

Please specify the full genotype of the mice. There are C57BL6j and C57BL6ej available at Jackson labs. The j version has a large-scale deletion in a gene encoding an important mitochondrial protein, NNT. This is not the case in the ej mice.

We now specified the full genotype of the mice used. In our work, we used C57BL6ej line, which has complete mitochondrial protein components. Please see amendment in section of “Experimental animals” in materials and methods.

Line 336: For SEM images, 512 x 512 seems to refer to the number of pixels per image rather than the pixel size. The values given on line 338 are also unclear. Please correct and specify the resolution in nanometers.

The numbers of pixels are 8928×8928 and 12288×12288 , and the resolution (pixel size) is $5 \text{ nm} \times 5 \text{ nm} \times 50 \text{ nm}$. This information is now provided in the Methods section.

Quantification of mitochondrial connectivity: please describe the method used with a sufficient level of details so the procedures could be independently replicated.

The method we used for mitochondrial segmentation and reconstruction has been described in full details in revised Methods section, under the subtitle of “EM images collection, alignment, and reconstruction”. Thank you!

Reference

- 1 Cichon, J. & Gan, W.-B. Branch-specific dendritic Ca^{2+} spikes cause persistent synaptic plasticity. *Nature* **520**, 180-185 (2015).

- 2 Morris, R. & Hollenbeck, P. J. T. J. o. c. b. Axonal transport of mitochondria along microtubules and F-actin in living vertebrate neurons. **131**, 1315-1326 (1995).
- 3 Misgeld, T., Kerschensteiner, M., Bareyre, F. M., Burgess, R. W. & Lichtman, J. W. J. N. m. Imaging axonal transport of mitochondria in vivo. **4**, 559 (2007).

Reviewers' Comments:

Reviewer #1:

Remarks to the Author:

Lin et al have resubmitted their manuscript on the coupling of cytosolic and mitochondrial calcium transients in cortical neurons in vivo. While – as originally pointed out – I see substantial merits in this study, i.e. in the effects of physiological activity or CaMK that it describes, the revisions have not sufficiently resolved my original concerns to recommend publication wholeheartedly. I am OK with the authors' decision to keep the EM (albeit it is not really used and the 'nanotunneling' adds nothing to explain the main topics of the study) and I appreciate the removal of the 'coupling potency' concept. However, I remain reserved as to how much of the reported unfaithful coupling at baseline could simply be related to imperfect measurements rather than true biology, and whether 'probabilistic' appropriately describes it. In this regards, the new in vitro data do not resolve my concerns as they are reported in a very cursory fashion and the addition of 'preliminary observations' regarding metabolic effects does not really seem suitable for a final paper revision. Throughout the study, I am missing a careful methodological analysis that would show, which aspects of the measurements can be taken as reliable description of how cytosolic and mitochondrial calcium transients are coupled in vivo in awake behaving animals, as opposed to reflect limitations of the measurements. With this I mean discussions of aspects such as signal-to-noise effects, sensor non-linearity and affinity, heterogeneity of the observed signals in cytoplasm and mitochondria (where many influences, some controlled, others not, are integrated) etc. This in the end leads to the use of the term 'probabilistic', which I read to mean something biological; perhaps the authors simply mean that their measurements or analyses might not have been precise or broad enough to find the relevant parameters. But then this has to be made clear, especially as the authors put their observations in contrast to prior in vitro observations that reported a more reliable coupling ('Together, these results suggest a weak, probabilistic coupling of $[Ca^{2+}]_{mito}$ to $[Ca^{2+}]_{cyto}$ in vivo, in contrast to previous observations in vitro 36-40.'). The latter sentence also seems outdated, now that the authors also claim 'Thus, in vitro experiments reproduced many salient features of the $[Ca^{2+}]_{mito}$ -to- $[Ca^{2+}]_{cyto}$ coupling in vivo.' – here an explanation for the diverging results would need to be provided.

In summary, while I still find this study conceptually interesting, the results regarding the effects of physiological activity and CaMK on coupling solid, and the question that it tries to answer important, the revised manuscript does not address all my original concerns in full, mostly because the paper does not delineate very well, what can be concluded from the reported experiments with certainty and what remains preliminary or tentative given the available methods.

Minor points:

Line 47: Remove 'the' before 'long term potentiation'

Line 48: Should be 'in dendrites'

Line 53: 'Despite these progresses' – unclear in reference and grammatically wrong; whole sentence needs revision, as it reads awkward – 'neuronal $[Ca^{2+}]_{mito}$ dynamics ... remain largely unknown.' is not correct.

Line 58: 'spatially synchronous' – there is no such thing, 'synchronous' refers to time.

Line 68: Better cite data from tissue EM, e.g. from K. Harris's work – the mitoflash measurements are likely underestimates, explaining the discrepancy to the values reported later.

Line 78: 'mitochondria in somas were manifested as short ovoid fragments' – revise; 'manifested as' reads awkward; 'fragments' suggests a previous 'intact' state

Line 82: 'in the whole cell body' – redundant and confusing

Line 129: What are 'Parametric measurements'?

Line 135: 'On the contrary' – wrong use of term, perhaps 'in contrast'?

Line 139: 'non-beat-to-beat' – what is beating here? This is a bit confusing, as it is not clear what is meant and whether the metaphor is from heart physiology or music.

Ref 24: Incomplete

Ref 28: Incomplete

Ref 29: Incomplete

Ref 30: Incomplete

Ref 32: Incomplete

Ref 33: Incorrect author list and incomplete

Reviewer #2:

Remarks to the Author:

The manuscript is substantially improved and is a good contribution to the field.

I would recommend adding the data on the mitochondrial modulation included on p.7 of the response to reviews.

Annotating the videos with labels to denote what each channel is (green, red) and to highlight specific cellular events of interest may help the reader.

Martin Picard

Reviewer #1 (Remarks to the Author):

Lin et al have resubmitted their manuscript on the coupling of cytosolic and mitochondrial calcium transients in cortical neurons in vivo. While – as originally pointed out – I see substantial merits in this study, i.e. in the effects of physiological activity or CaMK that it describes, the revisions have not sufficiently resolved my original concerns to recommend publication wholeheartedly. I am OK with the authors' decision to keep the EM (albeit it is not really used and the 'nanotunneling' adds nothing to explain the main topics of the study) and I appreciate the removal of the 'coupling potency' concept. However, I remain reserved as to how much of the reported unfaithful coupling at baseline could simply be related to imperfect measurements rather than true biology, and whether 'probabilistic' appropriately describes it. In this regards, the new in vitro data do not resolve my concerns as they are reported in a very cursory fashion and the addition of

'preliminary observations' regarding metabolic effects does not really seem suitable for a final paper revision. Throughout the study, I am missing a careful methodological analysis that would show, which aspects of the measurements can be taken as reliable description of how cytosolic and mitochondrial calcium transients are coupled in vivo in awake behaving animals, as opposed to reflect limitations of the measurements. With this I mean discussions of aspects such as signal-to-noise effects, sensor non-linearity and affinity, heterogeneity of the observed signals in cytoplasm and mitochondria (where many influences, some controlled, others not, are integrated) etc. This in the end leads to the use of the term 'probabilistic', which I read to mean something biological; perhaps the authors simply mean that their measurements or analyses might not have been precise or broad enough to find the relevant parameters. But then this has to be made clear, especially as the authors put their observations in contrast to prior in vitro observations that reported a more reliable coupling ('Together, these results suggest a weak, probabilistic coupling of $[Ca^{2+}]_{mito}$ to $[Ca^{2+}]_{cyto}$ in vivo, in contrast to previous observations in vitro 36-40.'). The latter sentence also seems outdated, now that the authors also claim 'Thus, in vitro experiments reproduced many salient features of the $[Ca^{2+}]_{mito}$ -to- $[Ca^{2+}]_{cyto}$ coupling in vivo.' – here an explanation for the diverging results would need to be provided.

In summary, while I still find this study conceptually interesting, the results regarding the effects of physiological activity and CaMK on coupling solid, and the question that it tries to answer important, the revised manuscript does not address all my original concerns in full, mostly because the paper does not delineate very well, what can be concluded from the reported experiments with certainty and what remains preliminary or tentative given the available methods.

We wholeheartedly appreciate the reviewer's further comments and suggestions. In response to your concerns centered around "a careful methodological analysis that would show, which aspects of the measurements can be taken as reliable description of how cytosolic and mitochondrial calcium transients are coupled in vivo in awake

behaving animals, as opposed to reflect limitations of the measurements”, we have included a new **Discussion** paragraph as the following:

“For the quantification of $[Ca^{2+}]_{mito}$ -to- $[Ca^{2+}]_{cyto}$ coupling, it is critical to ensure sensitive measurements for both $[Ca^{2+}]_{mito}$ and $[Ca^{2+}]_{cyto}$ events. The detection of $[Ca^{2+}]_{mito}$ response is challenged by multiple factors, including indicator affinity, non-linearity, and dynamic range (fluorescence for Ca^{2+} -free and bound species), as well as the basal and peak Ca^{2+} levels, and mitochondrial fractional volume. We opted to use GCaMP6f and jRGECO1a, for dual-color measurement of $[Ca^{2+}]_{mito}$ and $[Ca^{2+}]_{cyto}$ because GCaMP6 exhibits higher sensitivity in detecting $[Ca^{2+}]_{mito}$ than jRGECO1a^{1,2}. Even though alkalizing pH environment (~8.0 in mitochondrial matrix) would increase GFP fluorescence in a deprotonation-dependent and Ca^{2+} -independent manner³, the dynamic range of GCaMP6f should be 3-4 times higher than that of jRGECO1a². Despite technical limitations and uncertainties with the measurement of $[Ca^{2+}]$ in different cellular compartments, several lines of evidence suggest that the unfaithful $[Ca^{2+}]_{mito}$ coupling to $[Ca^{2+}]_{cyto}$ is likely a genuine physiological phenomenon. First, discrete $[Ca^{2+}]_{mito}$ transients, with their sudden and abrupt rises and distinctive long durations, were clearly discernible either from the baseline or on top of an ongoing $[Ca^{2+}]_{mito}$ transient. Even though $[Ca^{2+}]_{mito}$ transients were detected with a more sensitive indicator than $[Ca^{2+}]_{cyto}$, the frequency of $[Ca^{2+}]_{mito}$ events was much lower than that of $[Ca^{2+}]_{cyto}$. Second, CaMKII inhibition did not affect $[Ca^{2+}]_{cyto}$, but reduced the coupling fidelity between $[Ca^{2+}]_{cyto}$ and $[Ca^{2+}]_{mito}$, suggesting the involvement of CaMKII activity in the coupling process. Third, consistent with the involvement of biochemical signaling between $[Ca^{2+}]_{cyto}$ and $[Ca^{2+}]_{mito}$, conspicuous, yet highly variable latencies up to a few seconds were found in a significant portion of these coupled events. This observation further suggests that the onset of a $[Ca^{2+}]_{mito}$ transient could reflect the probabilistic gating of some fast mitochondrial Ca^{2+} uptake mechanism. In addition, our *in vitro* data show that such inherent probabilistic nature of the $[Ca^{2+}]_{mito}$ -to- $[Ca^{2+}]_{cyto}$ coupling could be masked in cultured neurons by strong electrical field stimulation used in previous studies⁴⁻¹⁰.”

By “probabilistic”, we mean that the coupling is gated by the fast mitochondrial calcium uptake mechanism (with mitochondrial calcium uniporter/MCU in mind), via a Ca^{2+} /CaMKII-dependent mechanism. In the Discussion, we stated “**we therefore propose that the triggering of cell-wide $[Ca^{2+}]_{mito}$ transient reflects cooperative opening of MCUs in the mitochondrial network, via a Ca^{2+} /CaMKII-dependent mechanism. In this scenario, uncoupling between $[Ca^{2+}]_{mito}$ and $[Ca^{2+}]_{cyto}$ may reflect that MCUs are not in their open state.**”

As the reviewer pointed out, the sentence “Together, these results suggest a weak, probabilistic coupling of $[Ca^{2+}]_{mito}$ to $[Ca^{2+}]_{cyto}$ *in vivo*, in contrast to previous observations *in vitro*.” was confusing. We have now modified this sentence to

"Together, these results suggest a weak, probabilistic coupling of $[Ca^{2+}]_{mito}$ to $[Ca^{2+}]_{cyto}$, particularly when neurons are not strongly activated *in vivo* and *in vitro*".

We would like to thank the reviewer again for helping us improve the manuscript. We hope the reviewer would be satisfied with the revision we have made above.

Minor points:

Line 47: Remove 'the' before 'long term potentiation'

We now corrected the term per your suggestion.

Line 48: Should be 'in dendrites'

This grammar mistake is corrected now.

Line 53: 'Despite these progresses' – unclear in reference and grammatically wrong; whole sentence needs revision, as it reads awkward – 'neuronal $[Ca^{2+}]_{mito}$ dynamics ... remain largely unknown.' is not correct.

We revised the sentence and it now reads: "Despite these *in vitro* research progresses, neuronal $[Ca^{2+}]_{mito}$ dynamics and its physiological regulation in the brain of awake behaving mammals remain largely unknown."

Line 58: 'spatially synchronous' – there is no such thing, 'synchronous' refers to time.

Thank you for pointing this out. We now rephrased it as "spatially coordinated"

Line 68: Better cite data from tissue EM, e.g. from K. Harris's work – the mitoflash measurements are likely underestimates, explaining the discrepancy to the values reported later.

This sentence is now deleted considering the limitations of optical measurement of mitochondrial length.

Line 78: 'mitochondria in somas were manifested as short ovoid fragments' – revise; 'manifested as' reads awkward; 'fragments' suggests a previous 'intact' state

This sentence is now revised. It reads, "mitochondria in somas were short ovoids..."

Line 82: 'in the whole cell body' – redundant and confusing

We now deleted the redundant words.

Line 129: What are 'Parametric measurements'?

Thank you for pointing out the error. We have now changed to "parametric analysis".

Line 135: 'On the contrary' – wrong use of term, perhaps 'in contrast'?

We now corrected the wording per your suggestion. Thank you.

Line 139: 'non-beat-to-beat' – what is beating here? This is a bit confusing, as it is not

clear what is meant and whether the metaphor is from heart physiology or music.
We now called it “unfaithful” coupling and hope this is ok.

Ref 24: Incomplete

Ref 28: Incomplete

Ref 29: Incomplete

Ref 30: Incomplete

Ref 32: Incomplete

Ref 33: Incorrect author list and incomplete

Thank you for pointing out our oversight. They have all been updated in the right format now.

Reviewer #2 (Remarks to the Author):

The manuscript is substantially improved and is a good contribution to the field.

I would recommend adding the data on the mitochondrial modulation included on p.7 of the response to reviews.

Annotating the videos with labels to denote what each channel is (green, red) and to highlight specific cellular events of interest may help the reader.

Martin Picard

We thank the reviewer for the strong support. Since the figure on the mitochondrial modulation in the previous response to the reviewer will be online published together with the paper in accordance with journal's policy, we hope you agree not to add it to the supplementary information.

References

- 1 Chen, T.-W. *et al.* Ultrasensitive fluorescent proteins for imaging neuronal activity. *Nature* **499**, 295 (2013).
- 2 Dana, H. *et al.* Sensitive red protein calcium indicators for imaging neural activity. *Elife* **5**, e12727 (2016).
- 3 Tsien, R. Y. THE GREEN FLUORESCENT PROTEIN. *Annual review of biochemistry* **67**, 509-544, doi:10.1146/annurev.biochem.67.1.509 (1998).
- 4 Rudolf, R., Mongillo, M., Magalhães, P. J. & Pozzan, T. In vivo monitoring of Ca²⁺ uptake into mitochondria of mouse skeletal muscle during contraction. *The Journal of cell biology* **166**, 527-536 (2004).
- 5 Poburko, D., Santo-Domingo, J. & Demaurex, N. Dynamic regulation of the mitochondrial proton gradient during cytosolic calcium elevations. *Journal of Biological Chemistry* **286**,

- 11672-11684 (2011).
- 6 Núñez, L. *et al.* Bioluminescence imaging of mitochondrial Ca²⁺ dynamics in soma and neurites of individual adult mouse sympathetic neurons. *The Journal of physiology* **580**, 385-395 (2007).
- 7 Hajnóczky, G., Robb-Gaspers, L. D., Seitz, M. B. & Thomas, A. P. Decoding of cytosolic calcium oscillations in the mitochondria. *Cell* **82**, 415-424 (1995).
- 8 Pacher, P., Thomas, A. P. & Hajnóczky, G. Ca²⁺ marks: miniature calcium signals in single mitochondria driven by ryanodine receptors. *Proceedings of the National Academy of Sciences* **99**, 2380-2385 (2002).
- 9 Csordás, G. *et al.* Imaging interorganelle contacts and local calcium dynamics at the ER-mitochondrial interface. *Molecular cell* **39**, 121-132 (2010).
- 10 De Stefani, D., Raffaello, A., Teardo, E., Szabò, I. & Rizzuto, R. A forty-kilodalton protein of the inner membrane is the mitochondrial calcium uniporter. *Nature* **476**, 336 (2011).